# VARIANCE-DEPENDENT REGRET LOWER BOUNDS FOR CONTEXTUAL BANDITS

**Jiafan He**
Department of Computer Science
University of California, Los Angeles
Los Angeles, CA 90095
jiafanhe19@ucla.edu

**Quanquan Gu**
Department of Computer Science
University of California, Los Angeles
Los Angeles, CA 90095
qgu@cs.ucla.edu

## ABSTRACT

Variance-dependent regret bounds for linear contextual bandits, which improve upon the classical $\widetilde{O}(d\sqrt{K})$ regret bound to $\widetilde{O}(d\sqrt{\sum_{k=1}^{K}\sigma_k^2})$, where $d$ is the context dimension, $K$ is the number of rounds, and $\sigma_k^2$ is the noise variance in round $k$, has been widely studied in recent years. However, most existing works focus on the regret upper bounds instead of lower bounds. To our knowledge, the only lower bound is from Jia et al. (2024), which proved that for any eluder dimension $d_{\mathbf{elu}}$ and total variance budget $\Lambda$, there exists an instance with $\sum_{k=1}^{K}\sigma_k^2 \leq \Lambda$ for which any algorithm incurs a variance-dependent lower bound of $\Omega(\sqrt{d_{\mathbf{elu}}\Lambda})$. However, this lower bound has a $\sqrt{d}$ gap with existing upper bounds. Moreover, it only considers a fixed total variance budget $\Lambda$ and does not apply to a general variance sequence $\{\sigma_1^2, \dots, \sigma_K^2\}$. In this paper, to overcome the limitations of Jia et al. (2024), we consider the general variance sequence under two settings. For a prefixed sequence, where the entire variance sequence is revealed to the learner at the beginning of the learning process, we establish a variance-dependent lower bound of $\Omega(d\sqrt{\sum_{k=1}^{K}\sigma_k^2/\log K})$ for linear contextual bandits. For an adaptive sequence, where an adversary can generate the variance $\sigma_k^2$ in each round $k$ based on historical observations, we show that when the adversary must generate $\sigma_k^2$ before observing the decision set $\mathcal{D}_k$, a similar lower bound of $\Omega(d\sqrt{\sum_{k=1}^{K}\sigma_k^2/\log^6(dK)})$ holds. In both settings, our results match the upper bounds of the SAVE algorithm (Zhao et al., 2023) up to logarithmic factors. Furthermore, if the adversary can generate the variance $\sigma_k$ after observing the decision set $\mathcal{D}_k$, we construct a counter-example showing that it is impossible to construct a variance-dependent lower bound if the adversary properly selects variances in collaboration with the learner. Our lower bound proofs use a novel peeling technique that groups rounds by variance magnitude. For each group, we construct separate instances and assign the learner distinct decision sets. We believe this proof technique may be of independent interest.

## 1 INTRODUCTION

We consider the linear contextual bandit problem, where each arm is represented by a feature vector and the expected reward is a linear function of this feature vector with an unknown parameter vector. Numerous studies have developed algorithms achieving optimal regret bounds for linear bandits (Chu et al., 2011; Abbasi-Yadkori et al., 2011). However, while these works establish minimax-optimal regret bounds in the worst-case, they do not exploit additional problem-dependent structures. Our work focuses on incorporating reward variance information into the analysis, building upon a line of research studying variance-dependent regret bounds for linear bandits (Zhou et al., 2021; Zhang et al., 2021; Zhou & Gu, 2022; Zhao et al., 2022; Kim et al., 2022; Zhao et al., 2023) and general function approximation (Jia et al., 2024), which includes linear bandits as a special case. Notably, Zhao et al. (2023) established a near-optimal regret guarantee without requiring prior knowledge of the variances:

**Theorem 1.1** (Theorem 2.3, Zhao et al. 2023). For any linear contextual bandit problem, the regret of the SAVE algorithm in the first $K$ rounds is upper bounded by:

$$\text{Regret}(K) \leq \widetilde{O}\Big(d\sqrt{\sum_{k=1}^{K} \sigma_k^2} + d\Big),$$

where $d$ is the dimension and $\sigma_k^2$ is the noise variance of the selected action in round $k$.

However, most of these works have focused on developing algorithms with regret upper bound guarantees, while variance-dependent lower bounds remain understudied. The only exception is Jia et al. (2024), which focuses on general function classes with finite eluder dimension $d_{\mathbf{elu}}$ and provides the following variance-dependent lower bound:

**Theorem 1.2** (Theorem 5.1, Jia et al. 2024). For any dimension $d \geq 2$, action space size $A$, number of rounds $K \geq 2$, and total variance budget $\Lambda \in [0, K]$, there exists a contextual bandit problem with eluder dimension $d_{\mathbf{elu}} = d$, action space size $A$, and an adversarial sequence of variances satisfying $\sum_{k=1}^{K} \sigma_k^2 \leq \Lambda$ such that for any algorithm, the regret is lower bounded by:

$$\text{Regret}(K) \geq \Omega\big(\min(\sqrt{d\Lambda} + d, \sqrt{AK})\big).$$

When restricted to the linear bandit case, where $d \geq \sqrt{A}$, the above lower bound reduces to $\sqrt{d\Lambda}$, which has a gap of $\sqrt{d}$ factor compared with the upper bound in Zhao et al. (2023). Moreover, Jia et al. (2024) only considers instances with a fixed budget $\Lambda$ and relies on carefully designed variance sequences $\{\sigma_1^2, \sigma_2^2, \ldots, \sigma_K^2\}$, failing to provide lower bounds for general variance sequences. Therefore, an open question arises:

*Can we prove variance-dependent regret lower bounds for general variance sequences?*

## 1.1 OUR CONTRIBUTIONS

In this paper, we answer this question affirmatively by constructing hard-to-learn instances in several different settings. For any prefixed sequence $\{\sigma_1^2, \ldots, \sigma_K^2\}$, we achieve a $\widetilde{\Omega}(d\sqrt{\sum_{k=1}^{K} \sigma_k^2})$ variance-dependent expected lower bound, which matches the upper bound in Zhao et al. (2023) up to logarithmic factors and demonstrates its optimality. For general adaptive variance sequences where a weak adversary (potentially collaborating with the learner) can generate variance $\sigma_k^2$ in each round $k$ based on historical observations, our instance provides a high-probability lower bound of $\widetilde{\Omega}(d\sqrt{\sum_{k=1}^{K} \sigma_k^2})$, which also matches the upper bound in Zhao et al. (2023) up to logarithmic factors. To the best of our knowledge, this is the first high-probability lower bound for linear contextual bandit.

Our construction and analysis rely on the following new techniques:

- A peeling technique for prefixed variance sequences that divides rounds into groups based on variance magnitude. Through orthogonal decision set construction, each group only interacts with its corresponding parameters, allowing us to establish separate lower bounds for different variance scales and combine them effectively.

- A multi-instance framework that handles unknown group sizes in the adaptive setting. For each variance group, we maintain multiple instances designed for different possible intervals of round numbers and assign the learner to these instances in a cyclic manner, ensuring uniform visits across instances and guaranteeing the visiting times of one instance matches its designed interval.

- A high-probability lower bound that handles adaptive group sizes through a union bound. We first convert expected regret bounds to constant-probability bounds through careful variance control and auxiliary algorithms, then boost these to high-probability bounds by creating multiple independent instances.

Furthermore, we also study the setting with a strong adversary that can generate the variance $\sigma_k$ after observing the decision set $\mathcal{D}_k$. Under this scenario, we proposed a counter algorithm that can collaborate with the adversary by properly selecting variance, achieving an $O(d)$ regret even the total variance $\sum_{k=1}^{K} \sigma_k^2 = \Omega(K)$. This implies that it is impossible to derive a variance-dependent lower bound for general variance sequence with strong adversary. As a direct extension of this result, we also show that it is impossible to derive a variance-dependent lower bound for stochastic linear bandits, where the decision set is fixed even for a general prefixed variance sequence.

**Notation** We use lower case letters to denote scalars, and use lower and upper case bold face letters to denote vectors and matrices respectively. We denote by $[n]$ the set $\{1, \ldots, n\}$. For a vector

$\mathbf{x} \in \mathbb{R}^d$ and a positive semi-definite matrix $\mathbf{\Sigma} \in \mathbb{R}^{d \times d}$, we denote by $\|\mathbf{x}\|_2$ the vector's $\ell_2$ norm and by $\|\mathbf{x}\|_{\mathbf{\Sigma}} = \sqrt{\mathbf{x}^\top \mathbf{\Sigma} \mathbf{x}}$ the Mahalanobis norm. For two positive sequences $\{a_n\}$ and $\{b_n\}$ with $n = 1, 2, \ldots$, we write $a_n = O(b_n)$ if there exists an absolute constant $C > 0$ such that $a_n \leq Cb_n$ holds for all $n \geq 1$ and write $a_n = \Omega(b_n)$ if there exists an absolute constant $C > 0$ such that $a_n \geq Cb_n$ holds for all $n \geq 1$. We use $\widetilde{O}(\cdot)$ to further hide the polylogarithmic factors. We use $\mathbb{1}\{\cdot\}$ to denote the indicator function.

## 2 RELATED WORK

**Heteroscedastic Linear Bandits.** For linear bandit problems, the worst-case regret has been widely studied (Auer, 2002; Dani et al., 2008; Li et al., 2010; Chu et al., 2011; Abbasi-Yadkori et al., 2011; Li et al., 2019), achieving $\widetilde{O}(\sqrt{K})$ bounds in the first $K$ rounds. Recently, a series of works has considered heteroscedastic variants where noise distributions vary across rounds. Kirschner & Krause (2018) first formally proposed a linear bandit model with heteroscedastic noise, assuming $\sigma_k$-sub-Gaussian noise in round $k \in [K]$. Subsequently, (Zhou et al., 2021; Zhang et al., 2021; Kim et al., 2022; Zhou & Gu, 2022; Dai et al.; Zhao et al., 2023; Jia et al., 2024) relaxed this to variance-based constraints where round $k$ has variance $\sigma_k^2$. Among these works, Zhou et al. (2021) and Zhou & Gu (2022) obtained near-optimal regret guarantees of $\widetilde{O}(d\sqrt{\sum_{k=1}^K \sigma_k^2})$, but required knowledge of $\sigma_k$ after observing the reward in round $k$. In contrast, Zhang et al. (2021); Kim et al. (2022) handled unknown variances with computationally inefficient algorithms, achieving a weaker $\widetilde{O}(\text{poly}(d)\sqrt{\sum_{k=1}^K \sigma_k^2})$ bound. Recently, Zhao et al. (2023) improved upon these results with an efficient algorithm (SAVE) achieving the near-optimal $\widetilde{O}(d\sqrt{\sum_{k=1}^K \sigma_k^2})$ bound without requiring variance knowledge. Beyond standard linear bandits, two directions have been explored. Dai et al. studied heteroscedastic sparse linear bandits, providing a framework to convert standard algorithms to the sparse setting. In a different direction, Jia et al. (2024) extended the analysis to contextual bandits with general function classes having finite eluder dimension, which includes linear bandits as a special case, and achieved a variance-dependent regret upper bounds.

**Lower Bounds for Linear Contextual Bandits.** For linear contextual bandit problems, several works (Dani et al., 2008; Chu et al., 2011; Li et al., 2019) have established theoretical lower bounds to illustrate the fundamental difficulty in learning process. For linear bandits with finite action sets, Chu et al. (2011) established an $\widetilde{\Omega}(\sqrt{dK})$ lower bound, matching the upper bound up to logarithmic factors in the action set size and number of rounds $K$. For general stochastic linear bandits, Dani et al. (2008) constructed an instance with $2^{\Omega(d)}$ actions and obtained an $\Omega(d\sqrt{K})$ lower bound. Later, Li et al. (2019) focused on linear contextual bandits, where the decision set can vary across rounds, and provided an $\Omega(d\sqrt{K \log K})$ lower bound. However, all these works only focus on worst-case regret bounds and do not consider the heteroscedastic variance information. The only exception is Jia et al. (2024), which provided an $\Omega(\sqrt{d\Lambda})$ variance-dependent lower bound for a fixed total variance budget $\Lambda$. Nevertheless, this work cannot handle general variance sequences and leaves open the question of variance-dependent lower bounds in the general setting.

**Variance-dependent Bounds for Multi-armed Bandits.** Auer et al. (2002) studied a multi-armed bandit problem in which the rewards are normally distributed with unknown mean and variance, and proposed the UCB1-NORMAL algorithm, which achieves a variance-dependent regret bound of $\widetilde{O}(\sum_{i=1, \Delta_i \neq 0}^n \frac{\sigma_i^2}{\Delta_i} + \Delta_i) \log K)$. Here $\sigma_i^2$ is the variance of the reward for the $i$-th arm, $\Delta_i$ is the suboptimality gap between $i$-th arm and the best arm, $n$ is the number of arms, and $K$ is the number of rounds. Audibert et al. (2009) considered a multi-armed bandit (MAB) problem with bounded reward (by $b > 0$) and unknown mean and variance. They proposed the UCB-V algorithm that achieves a variance-dependent regret bound of $\widetilde{O}(\sum_{i=1, \Delta_i \neq 0}^n \frac{\sigma_i^2}{\Delta_i} + b) \log K)$. They also established a matching lower bound. Variance-dependent regret bounds have also been established for best-arm identification problem (Audibert & Bubeck, 2010) in multi-armed stochastic bandits. For example, Lu et al. (2021) studied the best-arm identification problem in stochastic multi-armed bandits, and proved a variance dependent lower bound of $\widetilde{\Omega}(\sum_{i=1, \Delta_i \neq 0}^n \frac{\sigma_i^2}{\Delta_i^2} + \frac{1}{\Delta_i})$. They also proposed an algorithm that achieves a nearly matching upper bound. Lalitha et al. (2023) studied fixed-budget best-arm identification with heterogeneous reward variances. It is worth noting that these variance-dependent regret results for MAB rely on the assumption that the arms are fixed. Consequently, the sub-optimality gap $\Delta_i$ and the variance $\sigma_i^2$ are assumed to remain constant across all rounds.

In sharp contrast, our focus is on the linear contextual bandits, where the decision set $\mathcal{D}_k$ changes adaptively. This change depends on the history of actions and rewards, meaning the set of available arms (and even the size of the action set) is not fixed.

## 3 PRELIMINARIES

In this work, we consider the heteroscedastic linear contextual bandit (Zhou et al., 2021; Zhang et al., 2021), where the noise variance varies across rounds. Let $K$ be the total number of rounds. In each round $k \in [K]$, the interaction between the learner and the environment proceeds as follows:

1. The environment generates an arbitrary decision set $\mathcal{D}_k \subseteq \mathbb{R}^d$, where each element represents a feasible action that can be selected by the learner;

2. The learner observes $\mathcal{D}_k$ and selects $\mathbf{x}_k \in \mathcal{D}_k$;

3. The environment generates the stochastic noise $\epsilon_k$ and reveals the stochastic reward $r_k = \langle \boldsymbol{\mu}, \mathbf{x}_k \rangle + \epsilon_k$ to the learner, where $\boldsymbol{\mu} \in \mathbb{R}^d$ is the unknown weight vector for the underlying linear reward function.

Without loss of generality, we assume the random noise $\epsilon_k$ in each round $k$ satisfies:

$$\mathbb{P}(|\epsilon_k| \le R) = 1, \quad \mathbb{E}[\epsilon_k | \mathbf{x}_{1:k}, \epsilon_{1:k-1}] = 0, \quad \mathbb{E}[\epsilon_k^2 | \mathbf{x}_{1:k}, \epsilon_{1:k-1}] = \sigma_k^2 \le 1, \forall k \in [K] \quad (3.1)$$

For any algorithm Alg and linear bandit instance $\mathcal{M}$, the cumulative regret is defined as follows:

$$\text{Regret}_{\text{Alg}}(K, \mathcal{M}) = \sum_{k \in [K]} \langle \mathbf{x}_k^*, \boldsymbol{\mu} \rangle - \langle \mathbf{x}_k, \boldsymbol{\mu} \rangle, \quad \text{where } \mathbf{x}_k^* = \arg\max_{\mathbf{x} \in \mathcal{D}_k} \langle \mathbf{x}, \boldsymbol{\mu} \rangle. \quad (3.2)$$

For simplicity, we may omit the subscripts Alg and/or $\mathcal{M}$ when there is no ambiguity. Additionally, with a slight abuse of notation, we may use $\sigma_k$ to represent the variance $\sigma_k^2$ (which is originally the standard deviation) when there is no ambiguity. In this work, we focus on providing variance-dependent lower bounds for the regret based on the variances sequence $\{\sigma_1, ..., \sigma_K\}$. We consider two settings for the variance sequence $\{\sigma_1, \ldots, \sigma_K\}$:

- **Prefixed Sequence**: The variance sequence is revealed to the learner at the beginning of the learning process.

- **Adaptive Sequence**: An adversary (potentially collaborating with the learner) can generate the variance $\sigma_k$ in each round $k$ based on historical observations, with the learner receiving each variance at the beginning of the corresponding round. This setting can be further divided into two categories based on the power of the adversary:

  - **Weak Adversary**: The adversary must generate the variance $\sigma_k$ before observing the decision set $\mathcal{D}_k$.

  - **Strong Adversary**: The adversary can generate the variance $\sigma_k$ after observing the decision set $\mathcal{D}_k$.

**Remark 3.1.** Unlike the typical adversarial setting focused on maximizing regret for a specific algorithm, our work uses the idea of an "adversary" to represent the environment's inherent ability to select the variance sequence. This "adversary" might even strategically choose variance levels ($\sigma_k$) based on the **past decision sets** $\mathcal{D}_k$ **observed so far**, potentially leading to variance levels that could temporarily improve the learner's performance or make the learning process appear easier. This seeming "cooperation," however, is ultimately aimed at exploring the fundamental lower bounds on regret that must hold for any learner in any environment. The key is that the variance is chosen **without direct knowledge of the true underlying patterns** $\mu$. When this "adversary" (our "strong adversary") can adjust the variance based on the learner's actions ($\mathcal{D}_k$), this strategic "cooperation," informed by past observations but blind to $\mu$, becomes more effective in probing the true limits of learnability and challenging our lower bound results.

## 4 VARIANCE-DEPENDENT LOWER BOUND WITH PREFIXED VARIANCE SEQUENCE

In this section, we consider the setting where the variance sequence $\{\sigma_1, \ldots, \sigma_K\}$ is prefixed and fully revealed to the learner at the beginning of the learning process.

### 4.1 MAIN RESULTS

We establish the following theorem for the variance-dependent lower bound.

**Theorem 4.1.** Let $d > 1$ and consider any prefixed sequence of variances $\{\sigma_1, ..., \sigma_K\}$ satisfying $\sum_{k=1}^{K} \sigma_k^2 \geq 1 + 384d^2$. For any algorithm Alg, there exists a hard linear contextual bandit instance such that each action $a \in \mathcal{D}_k$ in round $k$ has variance bounded by $\sigma_k$. For this instance, the expected regret of algorithm Alg over $K$ rounds is lower bounded by:

$$\mathbb{E}[\text{Regret}(K)] \geq \Omega\Big(d\sqrt{\sum_{i=1}^{K} \sigma_k^2}/(\log K)\Big).$$

**Remark 4.2.** For a prefixed sequence $\{\sigma_1, ..., \sigma_K\}$, Theorem 4.1 shows that any algorithm incurs a regret lower bounded of $\widetilde{\Omega}(d\sqrt{\sum_{k=1}^{K} \sigma_k^2})$, which matches the upper bound in Zhao et al. (2023) up to logarithmic factors. Compared to the lower bound in Jia et al. (2024), Theorem 4.1 focuses on the linear contextual bandit setting and achieves a $\sqrt{d}$ improvement over the standard linear bandit setting. It is also worth noting that the lower bound in Jia et al. (2024) only considers instances with a fixed total variance $\sum_{k=1}^{K} \sigma_k^2$, constructed by using constant variance in the early rounds and zero variance in later rounds. In comparison, Theorem 4.1 applies to any fixed variance sequence and is more flexible.

In Theorem 4.1, we require that the total variance is no less than $\Omega(d^2)$, which reduces to $K \geq \Omega(d^2)$ when all variances $\sigma_k = 1$. A similar requirement exists in standard linear bandits, since a trivial lower bound of $\Omega(K)$ always holds for any algorithm, and the lower bound of $\Omega(d\sqrt{K})$ can only be achieved when $K \geq \Omega(d^2)$. Furthermore, for general sequences of variances with total variance smaller than $O(d^2)$, a large number of rounds $K$ alone is not sufficient to establish the desired lower bound. The presence of early rounds with zero variance would increase the total number of rounds without affecting the fundamental complexity of the problem. This observation suggests that requiring total variance no less than $\Omega(d^2)$ (or other equivalent conditions) may be necessary for establishing the lower bound.

### 4.2 PROOF OVERVIEW OF THEOREM 4.1

In this subsection, we prove the variance-dependent lower bound in Theorem 4.1. We first start with a fixed variance threshold $\sigma$, and construct a class of hard-to-learn instances where actions are chosen from a hypercube action set $\mathcal{A} = \{-1, 1\}^d$, and for any action $\mathbf{a} \in \mathcal{A}$, the reward follows a scaled Bernoulli distribution $\sigma \cdot B(1/3 + \langle \boldsymbol{\mu}, \mathbf{a} \rangle)$, where $\Delta = 1/\sqrt{96K}$ and $\boldsymbol{\mu} \in \{-\Delta, \Delta\}^d$. In this setting, the variance for each action is upper bounded by $\sigma^2$, and these instances can be represented as a linear bandit problem with feature $(\sigma, \sigma \cdot \mathbf{a})$ and weight vector $\boldsymbol{\mu}' = (1/3, \boldsymbol{\mu})$. Based on these hard-to-learn instances, we have the following variance-dependent lower bound for the regret:

**Lemma 4.3.** For a fixed variance threshold $\sigma$ and any bandit algorithm Alg, if the weight vector $\boldsymbol{\mu} \in \{-\Delta, \Delta\}^d$ is uniformly random selected from $\{-\Delta, \Delta\}^d$, the variance in each round is bounded by $\sigma^2$, and the expected regret over $K \geq 1.5 \cdot d^2$ rounds is lower bounded by:

$$\mathbb{E}_{\boldsymbol{\mu}}[\text{Regret}(K)] \geq d\sqrt{K\sigma^2}/8\sqrt{6}.$$

**Remark 4.4.** Lemma 4.3 establishes a variance-dependent lower bound for the regret with a fixed variance threshold $\sigma$. When all variances are equal ($\sigma_1 = ... = \sigma_K = \sigma$), this bound matches the upper bound in Zhao et al. (2023) up to logarithmic factors. In addition, under this fixed-variance setting, this lemma provides a tighter logarithmic dependency on the number of rounds $K$ compared to Theorem 4.1, though it does not extend to dynamic variances.

Now, for any prefixed variance sequence $\{\sigma_1, ..., \sigma_K\}$, we divide the rounds into $L = \lceil \log_2 K \rceil + 1$ different groups based on the range of their variance as follows:

$$\mathcal{K}_0 = \{k : \sigma_k \leq 1/K\},$$
$$\mathcal{K}_i = \{k : 2^{i-1}/K < \sigma_k \leq 2^i/K\}, \quad \text{for } i = 1, \ldots, L-1.$$

For each group $\mathcal{K}_i$ with $i \in [L-1]$, we construct a bandit instance $\mathcal{M}_i$ with weight vector $\boldsymbol{\mu}_i$ following Lemma 4.3, where:

- the variance threshold is set to be $\sigma(i) = 2^{i-1}/K$;
- the number of rounds is $K_i = |\mathcal{K}_i|$;

- the dimension is $d_i = d/L$.

For group $\mathcal{K}_0$, we construct a different type of instance $\mathcal{M}_0$: a $d/L$-armed bandit, where one randomly chosen arm gives constant reward 1 while all other arms give reward 0. Note that this instance in $\mathcal{M}_0$ can be equivalently represented as a $d_0 = d/L$-dimensional linear bandit where actions are one-hot vectors $\mathbf{e}_i$.

The basic idea for the lower regret bound is to assign different orthogonal sub-instances based on the range of the variance $\sigma_k$ at the beginning of each round. This method ensures that each orthogonal instance will be learned with comparable variance, which makes it easier to derive a tighter lower regret bound. Finally, since the orthogonal instances cannot provide mutual information, the total regret can be decomposed into the summation of the regret accumulated in each sub-instance.

Based on these sub-instances, we create a combined linear bandit instance with dimension $d_0 + d_1 + ... + d_{L-1} = d$ with weight vector $\boldsymbol{\mu} = (\boldsymbol{\mu}_0, ..., \boldsymbol{\mu}_{L-1})$: At the beginning of each round $k$, if round $k$ belongs to group $\mathcal{K}_i$, then the learner receives the decision set $\mathcal{D}_k = \left\{ (\mathbf{0}_{d_0}, ..., \mathbf{0}_{d_{i-1}}, \mathbf{x}, \mathbf{0}_{d_{i+1}}, ..., \mathbf{0}_{d_{L-1}}) : \mathbf{x} \in \mathcal{A}_i \right\}$, where $\mathbf{0}_{d_j}$ corresponds to a zero vector with dimension $d_j$ and $\mathcal{A}_i$ is the action set in the bandit instance $\mathcal{M}_i$. Under this construction, for any round $k \in \mathcal{K}_i$, the reward in the combined instance coincides with that of sub-instance $\mathcal{M}_i$. Specifically, after the learner selects action $\mathbf{x}$, they receive a reward drawn from a scaled Bernoulli distribution with variance upper bounded by $\sigma^2(i) = \left(2^{i-1}/K\right)^2$ for $i \neq 0$, and variance 0 for $i = 0$. Note that in all groups, the variance is bounded by $\sigma_k^2$. With this construction in hand, we now proceed to prove the lower bound in Theorem 4.1.

**Remark 4.5** (Linear Contextual Bandits vs. Stochastic Linear Bandits). In the proof of Theorem 4.1, we heavily rely on assigning different decision sets to rounds in the contextual bandit environment. This approach, however, does not extend to stochastic linear bandit problems, where all rounds share the same decision set. To see this limitation, consider any prefixed variance sequence with $\sigma_1 = \cdots = \sigma_d = 0$. In this case, the learner can select canonical basis of the decision set in the first $d$ rounds. Since these rounds have zero variance, the learner learns the exact rewards for all actions in the decision set and incurs no regret in subsequent rounds, regardless of the values of $\sigma_{d+1}, \ldots, \sigma_K$. Consequently, it is impossible to establish an variance-aware lower bound of $\widetilde{\Omega}(d\sqrt{\sum_{k=1}^{K} \sigma_k^2})$ for stochastic linear bandits.

*Proof of Theorem 4.1.* Due to the orthogonal construction of decision sets across different groups $\mathcal{K}_i$, actions in group $\mathcal{K}_i$ provide no information about the weight vector $\boldsymbol{\mu}_j$ for $j \neq i$. Consequently, the total regret can be decomposed into the sum of regrets from each sub-instance. For each sub-instance $\mathcal{M}_i$ with $i \neq 0$, the regret is lower bounded by:

$$\mathbb{E}_{\boldsymbol{\mu}_i}\left[ \sum_{k \in \mathcal{K}_i} \max_{\mathbf{x} \in \mathcal{D}_k} \langle \boldsymbol{\mu}_i, \mathbf{x} \rangle - \langle \boldsymbol{\mu}_i, \mathbf{x}_k \rangle \right] \geq \mathbb{1}(K_i \geq 1.5d_i^2) \cdot \frac{d_i \sqrt{K_i \sigma^2(i)}}{8\sqrt{6}}$$

$$\geq \frac{d_i \sqrt{K_i \sigma^2(i)}}{8\sqrt{6}} - \frac{d_i \sqrt{1.5d_i^2 \cdot \sigma^2(i)}}{8\sqrt{6}}$$

$$\geq \frac{d_i \sqrt{\sum_{k \in \mathcal{K}_i} \sigma_k^2}}{16\sqrt{6}} - \frac{d_i^2 \cdot \sigma(i)}{16}, \tag{4.1}$$

where the first inequality follows from Lemma 4.3, the second inequality holds due to $\mathbb{1}(x \geq y)\sqrt{x} \geq \sqrt{x} - \sqrt{y}$, and the last inequality follows from the definition of group $\mathcal{K}_i$.

Taking a summation of (4.1) over all groups, the total regret can be lower bounded as follows:

$$\mathbb{E}_{\boldsymbol{\mu}}[\text{Regret}(K)] = \sum_{i=0}^{L-1} \mathbb{E}_{\boldsymbol{\mu}_i}\left[ \sum_{k \in \mathcal{K}_i} \max_{\mathbf{x} \in \mathcal{D}_k} \langle \boldsymbol{\mu}_i, \mathbf{x} \rangle - \langle \boldsymbol{\mu}_i, \mathbf{x}_k \rangle \right]$$

$$\geq \sum_{i=1}^{L-1} \frac{d_i \sqrt{\sum_{k \in \mathcal{K}_i} \sigma_k^2}}{16\sqrt{6}} - \frac{d_i^2 \cdot \sigma(i)}{16}$$

$$\geq \sum_{i=1}^{L-1} \frac{d \sqrt{\sum_{k \in \mathcal{K}_i} \sigma_k^2}}{16\sqrt{6}L} - \frac{d^2}{4L^2}$$

$$\geq \frac{d\sqrt{\sum_{i=1}^{L-1}\sum_{k\in\mathcal{K}_i}\sigma_k^2}}{16\sqrt{6}L} - \frac{d^2}{4L^2}, \tag{4.2}$$

where the first inequality follows from (4.1), the second inequality follows from the definition of variance threshold $\sigma(i)$ and dimension $d_i = d/L$, and the last inequality holds due to $\sum_i \sqrt{x_i} \geq \sqrt{\sum_i x_i}$. In addition, for the group $\mathcal{K}_0$, we have

$$\sum_{k\in\mathcal{K}_0}\sigma_k^2 \leq \sum_{k\in\mathcal{K}_0} 1/K \leq 1, \tag{4.3}$$

where the first inequality follows from the definition of group $\mathcal{K}_0$ and the second inequality follows from $|\mathcal{K}_0| \leq K$. Therefore, we have

$$\begin{aligned}
\mathbb{E}_{\boldsymbol{\mu}}[\mathrm{Regret}(K)] &\geq \frac{d\sqrt{\sum_{i=1}^{L-1}\sum_{k\in\mathcal{K}_i}\sigma_k^2}}{16\sqrt{6}L} - \frac{d^2}{4L^2} \\
&\geq \frac{d\sqrt{\sum_{k=1}^{K}\sigma_k^2 - 1}}{16\sqrt{6}L} - \frac{d^2}{4L^2} \\
&\geq \frac{d\sqrt{\sum_{k=1}^{K}\sigma_k^2 - 1}}{32\sqrt{6}L},
\end{aligned}$$

where the first inequality follows from (4.2), the second inequality follows from (4.3), and the last inequality follows from the fact that $\sum_{k=1}^{K}\sigma_k^2 \geq 1 + 384d^2$. This completes the proof. □

## 5 VARIANCE-DEPENDENT LOWER BOUNDS WITH ADAPTIVE VARIANCE SEQUENCE

In the previous section, we focused on the setting where the variance sequence is prefixed and revealed to the learner at the beginning of the learning process. In this section, we extend our analysis to the setting where the variance sequence can be adaptive based on historical observations, with the learner receiving the adaptive variance at the beginning of each round.

### 5.1 MAIN RESULTS

#### 5.1.1 WEAK ADVERSARY

We first describe the learning process and the mechanism of variance adaptation. In detail, the adaptive variance process proceeds as follows:

1. At the beginning of each round $k$, a (weak) adversary selects the variance level $\sigma_k$ based on the historical observations, including actions $\{a_1, \ldots, a_{k-1}\}$, rewards $\{r_1, \ldots, r_{k-1}\}$, and decision sets $\{\mathcal{D}_1, \mathcal{D}_2, \ldots, \mathcal{D}_{k-1}\}$. The adversary has access to all historical information but not to the underlying reward model parameters;

2. Given the selected variance level $\sigma_k$, we construct and assign a decision set $\mathcal{D}_k$ to the learner, where the variance of the reward for each action $a \in \mathcal{D}_k$ is bounded by $\sigma_k^2$;

3. The learner observes the decision set $\mathcal{D}_k$ and variance level $\sigma_k$, then determines an action $a_k$ from $\mathcal{D}_k$ based on its historical observations and current information. After selecting the action, the learner receives a reward $r_k$ with variance bounded by $\sigma_k^2$.

**Remark 5.1.** It is worth noting that our concept of adversary differs from the weak/strong adversary in Jia et al. (2024). Specifically, Jia et al. (2024) considers an adversary that attempts to hinder the learner's learning by allocating a fixed total variance budget $\sum_{k=1}^{K}\sigma_k^2 \leq \Lambda$ across rounds to maximize regret. In contrast, our work considers an adversary that attempts to break the lower bounds themselves by collaborating with the learner. To prevent such exploitation, we must restrict the adversary from knowing the weight vector of the underlying reward model. Without this restriction, the adversary could encode each entry $\mu_i$ of the weight vector $\boldsymbol{\mu}$ through the corresponding variance $\sigma_i = \mu_i$, allowing the learner to learn the weight vector after $d$ rounds.

Under this setting, we establish the following theorem for the variance-dependent lower bound.

**Theorem 5.2** (Weak Adversary). For any dimension $d > 1$, any adaptive sequence of variances $\{\sigma_1, \ldots, \sigma_K\}$ and any algorithm **Alg**, there exists a hard instance such that each action $a \in \mathcal{D}_k$ in round $k$ has variance bounded by $\sigma_k^2$. For this instance, if $\sum_{k=1}^{K} \sigma_k^2 \geq \Omega(d^2)$, then with probability at least $1 - 1/K$, the regret of **Alg** over $K$ rounds is lower bounded by:

$$\text{Regret}(K) \geq \Omega\Big(d\sqrt{\sum_{k=1}^{K} \sigma_k^2} / \log^6(dK)\Big).$$

**Remark 5.3.** Theorem 5.2 provides a high-probability lower bound of $\widetilde{\Omega}\big(d\sqrt{\sum_{k=1}^{K} \sigma_k^2}\big)$, which matches the upper bound in Zhao et al. (2023) up to logarithmic factors, albeit with looser logarithmic dependencies than Theorem 4.1 due to the adaptive nature of the variance sequence. Unlike the expected lower bound in Theorem 4.1, for adaptive variance sequences, the cumulative variance $\sum_{k=1}^{K} \sigma_k^2$ depends on the random process and observations. This dependence makes it challenging to establish an expected variance-dependent regret bound - a fundamental difficulty that does not arise for standard $d\sqrt{K}$-type lower bounds in linear contextual bandits (Dani et al., 2008; Chu et al., 2011; Lattimore & Szepesvári, 2018). To the best of our knowledge, our result provides the first high-probability lower bound for linear contextual bandits. In addition, our technique of converting an expected lower bound to a high-probability one is of independent interest and can potentially be used to derive high-probability lower bounds for a wider class of problems.

### 5.1.2 STRONG ADVERSARY

In Theorem 5.2, we require that for each round $k \in [K]$, all actions $\mathbf{x} \in \mathcal{D}_k$ share the same adaptive variance $\sigma_k$. This is more restrictive than the setting in Zhao et al. (2023), where the variance can differ across actions $\mathbf{x} \in \mathcal{D}_k$. However, extending our lower bound to action-dependent variances is fundamentally incompatible with our adaptive instance construction. The key difficulty is that, in our lower-bound construction, the decision set $\mathcal{D}_k$ is generated before the adversary chooses the variance $\sigma_k$, which prevents assigning specific variances to individual actions $\mathbf{x} \in \mathcal{D}_k$. Moreover, we now consider a strong adversary that can choose $\sigma_k$ after observing the decision set $\mathcal{D}_k$. The interaction between the learner and this strong adversary proceeds as follows:

1. At the beginning of each round $k$, we construct and assign a decision set $\mathcal{D}_k$ based on historical observations, including actions $\{a_1, \ldots, a_{k-1}\}$ and rewards $\{r_1, \ldots, r_{k-1}\}$;

2. Given the decision set $\mathcal{D}_k$ in round $k$, the strong adversary selects the variance level $\sigma_k$ for round $k$. The adversary has access to all historical information but not to the underlying reward model parameters;

3. The learner observes the decision set $\mathcal{D}_k$ and variance level $\sigma_k$, then determines an action $a_k$ from $\mathcal{D}_k$ based on its historical observations and current information. After selecting the action, the learner receives a reward $r_k$ with variance bounded by $\sigma_k^2$.

The following theorem shows that under this setting, the adversary could cooperate with the learner to break the lower bound.

**Theorem 5.4** (Strong Adversary). For any linear contextual bandit problem and number of rounds $K \geq 2d$, if we first provide the decision set $\mathcal{D}_k$ and then allow an adversary to choose the variance $\sigma_k$ based on the decision set $\mathcal{D}_k$, there exists one such type of adversary such that, there exists an algorithm whose regret in the first $K$ rounds is upper bounded by $\text{Regret}(K) \leq d$, where the total variance $\sum_{k=1}^{K} \sigma_k^2 \geq K/2$.

**Remark 5.5.** Theorem 5.4 highlights why Theorem 5.2 requires a weak adversary that set the variance sequence before seeing the learner's choices. If the adversary could see the decision set first, it could potentially choose variances that would invalidate our lower bound. This finding underscores that our construction is precise and pinpoints the exact condition under which the derived lower bound holds.

**Remark 5.6.** It is worth noting that Jia et al. (2024) also considered the case where the adversary assigns variances to actions after observing the decision set and action choice, and provided a variance-dependent lower bound. However, their analysis focuses on an adversary that allocates variance across rounds to maximize the regret. In contrast, our work considers an adversary that attempts to break these bounds, making it more challenging to establish lower bounds for general variance sequences. It is also worth noting that if the adversary's goal is to increase regret, choosing a prefixed sequence is a viable strategy. This case is already covered by our Theorem 4.1 for prefixed sequences, which provides a tighter lower bound than Theorem 5.2.

Theorem 5.4 suggests that it is impossible to derive a variance-dependent lower bound if the adversary can determine the variance $\sigma_k$ after observing the decision set $\mathcal{D}_k$, which further precludes establishing a lower bound when the adversary has the ability to assign action-dependent variances for each action $\mathbf{x} \in \mathcal{D}_k$ after observing the decision set $\mathcal{D}_k$. This result naturally extends to stochastic linear bandit problems, where the decision set $\mathcal{D}$ remains fixed across all rounds. In this case, since the adversary knows the decision set $\mathcal{D}_k = \mathcal{D}$ in advance, Theorem 5.4 directly implies:

**Corollary 5.7.** For any stochastic linear bandit problem with fixed decision set $\mathcal{D}$ and number of rounds $K \geq 2d$, there exists a prefixed sequence $\{\sigma_1, \ldots, \sigma_K\}$ such that there exists an algorithm whose regret in the first $K$ rounds is upper bounded by: $\text{Regret}_{\text{Alg}}(K) \leq d$, where the total variance $\sum_{k=1}^{K} \sigma_k^2 \geq K/2$.

## 5.2 Proof Sketch of Theorem 5.2

In this section, we provide the proof sketch of Theorem 5.2. Overall, the proof follows a similar structure as Theorem 4.1, where we divide the rounds into several groups based on their variance magnitude and create hard instances for each group. The key idea is to calculate individual regret bounds for each group and combine them for the final lower bound. However, there exist several challenges when dealing with adaptive variance sequences that require careful handling.

**Varying Size of Groups $\mathcal{K}_i$** As discussed in Section 4.2, for each group $\mathcal{K}_i$, we create individual instance $\mathcal{M}_i$ with fixed variance threshold $\sigma(i) = 2^{i-1}/K$ and establish a lower bound of $\widetilde{\Omega}(d_i\sqrt{\sigma^2(i)|\mathcal{K}_i|})$ on the expected regret. However, the construction of such instances relies on prior knowledge of the number of rounds $|\mathcal{K}_i|$, which can be calculated at the beginning for a prefixed variance sequence $\{\sigma_1, \ldots, \sigma_K\}$. In contrast, for general adaptive variance sequences, the number of rounds $|\mathcal{K}_i|$ is not known a priori and can even be a random variable, which creates a barrier in constructing these instances.

To address the unknown number of rounds $|\mathcal{K}_i|$, instead of constructing a single instance $\mathcal{M}_i$ for each group, we create $L$ instances $\mathcal{M}_{i,j}$, where $L = \lceil \log_2 K \rceil + 1$. Each instance $\mathcal{M}_{i,j}$ is designed for a specific range of round numbers, specifically $\mathcal{M}_{i,j}$ for $2^{j-1} \leq |\mathcal{K}_i| < 2^j$.

For each round $k$ in group $\mathcal{K}_i$, the learner receives a decision set $\mathcal{D}_i$ from one of the instances in $\{\mathcal{M}_{i,1}, \ldots, \mathcal{M}_{i,L}\}$ in a cyclic manner. Through this sequential assignment, the number of visits to each instance $\mathcal{M}_{i,j}$ is $|\mathcal{K}_i|/L$. Consequently, we expect that the instance $\mathcal{M}_{i,j}$ corresponding to the true range $2^{j-1} \leq |\mathcal{K}_i| < 2^j$ provides a lower bound of $\widetilde{\Omega}(d_i\sqrt{\sigma^2(i)|\mathcal{K}_i|}) = \widetilde{\Omega}(d_i\sqrt{\sigma^2(i) \cdot 2^j})$, which leads to the final lower bound of $\widetilde{\Omega}(d\sqrt{\sum_{k=1}^{K} \sigma_k^2})$.

**Converting Expected Lower Bound to High-Probability Lower Bound.** Another challenge is establishing the lower bound for the triggered instance $\mathcal{M}_{i,j}$ corresponding to the true range $2^{j-1} \leq |\mathcal{K}_i| < 2^j$. Traditional analysis of lower bounds in linear contextual bandits has focused on the expected regret. However, when dealing with adaptive variance sequences, this approach becomes insufficient as the adversary can dynamically adjust the variance sequence to break these bounds.

For instance, an adversary might continuously set $\sigma_k = 1$ until the lower bound of $\widetilde{\Omega}(d\sqrt{\sum_{i=1}^{k} \sigma_i^2})$ is violated at some round $k$, then switch to $\sigma_k = 0$ for all future rounds, causing the total variance sum $\sum_{k=1}^{K} \sigma_k^2$ to remain unchanged. In our construction, this means all rounds could fall into group $\mathcal{K}_L$, allowing the adversary to adaptively change the number of rounds between different intervals $2^{j-1} \leq |\mathcal{K}_L| < 2^j$. Since the failure of the lower bound in any single instance $\mathcal{M}_{L,j}$ leads to failure of the whole construction, an expected lower bound on regret cannot guarantee robust performance against adaptive sequences. This necessitates a stronger high-probability lower bound that holds uniformly for all instances.

Unfortunately, an expectation of $\widetilde{\Omega}(d_i\sqrt{\sigma^2(i)2^j})$ in instance $\mathcal{M}_{i,j}$ only implies a low-probability regret $\left(\text{Regret} \geq \widetilde{\Omega}(d_i\sqrt{\sigma^2(i)2^j})\right) \geq d_i \cdot 2^{-j/2}$, since the cumulative regret in $\mathcal{K}_i$ can be up to $\sigma(i) \cdot |\mathcal{K}_i|$ in our instance. To solve this problem, we introduce an auxiliary algorithm that automatically detects the cumulative regret and switches to the standard OFUL algorithm (Abbasi-Yadkori et al., 2011) if the cumulative regret is larger than $\Omega(d_i\sqrt{\sigma^2(i)2^j})$.[1] For this auxiliary algorithm, we can

---

[1] In general settings, detecting cumulative regret is impossible as the learner lacks prior knowledge of the optimal reward and variance. However, in our lower bound construction, all instances are randomly selected from instance classes sharing the same optimal reward and variance, which are known to the learner. This knowledge enables the construction of the auxiliary algorithm.

guarantee that the upper bound is at most $\widetilde{\Omega}(d_i\sqrt{\sigma^2(i)2^j})$ while maintaining the same probability of high regret as the original algorithm. Therefore, an expectation of $\widetilde{\Omega}(d_i\sqrt{\sigma^2(i)2^j})$ in instance $\mathcal{M}_{i,j}$ implies a constant-probability regret $\mathbb{P}\big(\text{Regret} \geq \widetilde{\Omega}(d_i\sqrt{\sigma^2(i)2^j})\big) = \Omega(1)$.

After constructing an instance with constant-probability lower bound, we boost this probability by creating $\Omega\big(\log^2(dK)\big)$ independent instances. When the learner encounters instance $\mathcal{M}_{i,j}$, it is assigned to one of these instances in a cyclic manner. Through this construction, with probability at least $1 - 1/\text{poly}(K)$, the final regret is lower bounded by $\text{Regret} \geq \widetilde{\Omega}(d_i\sqrt{\sigma^2(i)2^j})$.

**Remark 5.8.** Unlike previous lower bounds for linear bandit problems which focus on expected regret, to the best of our knowledge, our result provides the first high-probability lower bound for linear contextual bandits. It is worth noting that our construction requires separate decision sets across different rounds in the random assignment process. For stochastic linear bandits with a fixed decision set, we can only derive a constant-probability lower bound. Moreover, for a fixed decision set in stochastic linear bandit problem with covering number $\log\mathcal{N} \leq \widetilde{O}(d)$, an algorithm can randomly select one action from the covering set and perform this action in all rounds. In this case, there exists a probability of $1/\mathcal{N} = 1/\exp(d)$ to achieve zero regret, which precludes the possibility of establishing high-probability lower bounds for large round numbers $K$. More details about the high-probability lower bound can be found in Section 5.2.

## 6 Conclusion and Future Work

In this paper, we study variance-dependent lower bounds for linear contextual bandits in different settings. For both prefixed and adaptive variance sequences with weak adversary, we establish tight lower bounds matching the upper bounds in Zhao et al. (2023) up to logarithmic factors. We further demonstrate a fundamental limitation: when a strong adversary can select variances after observing decision sets, it becomes impossible to establish meaningful variance-dependent lower bounds. However, our work has focused exclusively on linear bandit settings, while Jia et al. (2024) has established variance-dependent lower bounds for general function approximation with a fixed total variance budget $\Lambda$. Therefore, we leave for future work the generalization of our analysis of general variance sequence to contextual bandits with general function approximation.

## Acknowledgment

We thank the anonymous reviewers and area chair for their helpful comments. JH and QG are supported in part by the National Science Foundation DMS-2323113 and IIS-2403400. JH is also supported by UCLA dissertation year fellowship. The views and conclusions contained in this paper are those of the authors and should not be interpreted as representing any funding agencies.

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

## LLM Usage

We used an LLM solely for grammatical and stylistic polishing of the manuscript. No research ideas or results were generated by the LLM. All technical content was written and verified by the authors.

## A  Experiments

In this section, we conduct experiments to show the difficulty of our construction of hard-to-learn instances.

### A.1  Experimental Setup

In this experiment, we follow the construction of hard-to-learn instances presented in the proof of Theorem 4.1, which breaks down the problem into several sub-instances.

**Model Parameters.** We consider a contextual linear bandit with total dimension $D = 10$, which we break down into two orthogonal sub-instances, $\mathcal{M}_1$ and $\mathcal{M}_2$, each with dimension $d_1 = d_2 = 5$. The environment is defined by the set of true parameter vectors $\mu = (\boldsymbol{\mu}_1, \boldsymbol{\mu}_2)$, where two fixed vectors, $\boldsymbol{\mu}_1$ and $\boldsymbol{\mu}_2$, each have non-zero entries drawn i.i.d. from $\mathcal{U}(0, 1)$ in only 5 dimensions.

**Variance Sequence** We consider a prefixed variance sequence over $K = 4000$ rounds. The variance sequence is piecewise, defined by an abrupt shift at $K_{\text{SWITCH}} = 2000$:

- **Low Variance ($\sigma_1 = 0.1$):** Used in the first 2000 rounds ($k \leq 2000$).
- **High Variance ($\sigma_2 = 1.0$):** Used in the subsequent 2000 rounds ($2000 < k \leq 4000$).

**Scenario Assignment** To illustrate the necessity of adaptively allocating different instances to the learner based on the variance level, we consider two scenarios for assigning the sub-instances ($\mathcal{M}_1$ or $\mathcal{M}_2$) to the learner:

1. **Piecewise Assignment (Hard-to-Learn):** The first 2000 rounds are assigned the instance $\mathcal{M}_1$, and the second 2000 rounds are assigned the instance $\mathcal{M}_2$. (Switch occurs at $T_{\text{SWITCH}}$).
2. **Alternating Assignment (Rapidly Switching):** The odd rounds are assigned the instance $\mathcal{M}_1$, and the even rounds are assigned the instance $\mathcal{M}_2$. (Switch occurs at each round).

**Decision Set** In each round $k$, the decision set $\mathcal{D}_k$ contains $N_{\text{arms}} = 32$ contexts. The base context entries are drawn i.i.d. from $\mathcal{U}(0, 1)$. This context set is masked such that contexts interact only with $\boldsymbol{\mu}_1$ when $\mathcal{M}_1$ is assigned, and only with $\boldsymbol{\mu}_2$ when $\mathcal{M}_2$ is assigned. Crucially, this orthogonal masking ensures that information gathered from one sub-instance cannot be transferred or used to estimate the parameter vector of the other sub-instance.

**Noise and Reward** For each round $k$, after the learner chooses an action $\mathbf{a} \in \mathcal{D}_k$, it receives a reward $r_k(\mathbf{a}) = \mathbf{a}^\top \boldsymbol{\mu}^* + \epsilon_k$, where the noise $\epsilon_k$ is drawn from the Gaussian distribution $\mathcal{N}(0, \sigma_i^2)$, with $\sigma_i$ determined by the prefixed variance sequence (i.e., $\sigma_i = 0.1$ for $k \leq 2000$ and $\sigma_i = 1.0$ for $k > 2000$).

### A.2  Results and Discussion

In the experiment, we evaluate the performance of two key algorithms:

- **OFUL** Abbasi-Yadkori et al. (2011): This provides a near-optimal **variance-independent** regret guarantee for the standard linear contextual bandit problem.
- **Weighted OFUL** Zhou et al. (2021): This provides a near-optimal **variance-dependent** regret guarantee for the linear contextual bandit problem, assuming the variance for each round is known to the learner.

We repeat each baseline algorithm for 100 times and plot their cumulative regrets with respect to the number of rounds in Figures 1 to 4.

#### A.2.1  Analysis of OFUL

As shown in Figure 1 and 2 (standard OFUL), the algorithm does not utilize the information regarding the variance level and constructs a similar confidence set size for the low-variance period and the later high-variance period, which leads to comparable regret in both periods. Also, for the alternating assignment (Figure 3), even though each sub-instance $\mathcal{M}_i$ alternates between low and high variance, OFUL fails to gain an advantage from the low-variance rounds and has a comparable total regret across all 4000 rounds.

### A.2.2 ANALYSIS OF WEIGHTED OFUL

The results for the Weighted OFUL algorithm (Figures 3 and 4), which utilizes a variance-based mechanism, show a totally different observation. For the Piecewise Assignment (Figure 3), Weighted OFUL utilizes the low variance in the first 2000 rounds and achieves a much lower regret in that initial period. However, since the information gathered is only for instance $\mathcal{M}_1$ and cannot transfer to the orthogonal instance $\mathcal{M}_2$, Weighted OFUL achieves a much higher regret in the last 2000 rounds due to the high variance.

In comparison, for the Alternating Assignment (Figure 4), each instance ($\mathcal{M}_1$ and $\mathcal{M}_2$) effectively goes through 1000 rounds with low variance and then 1000 rounds with high variance. Under this situation, Weighted OFUL can construct a tighter confidence set for both instances in the first 2000 rounds, which leads to a much smaller total regret over 4000 rounds. This result illustrates that the early stage with low variance can significantly speed up the learning process in the exploration stage and lead to a low total regret. The ability to adaptively assign the decision set based on the variance level (as we used in constructing the lower bound) can avoid the early exploration stage having much smaller variance than the later exploitation stage, thus successfully circumventing the limitation.

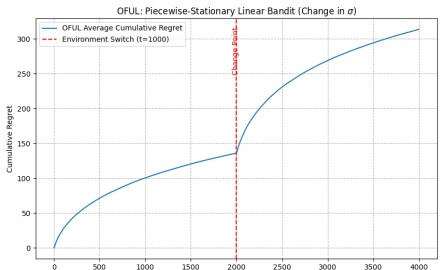

Figure 1: OFUL with Piecewise Assignment

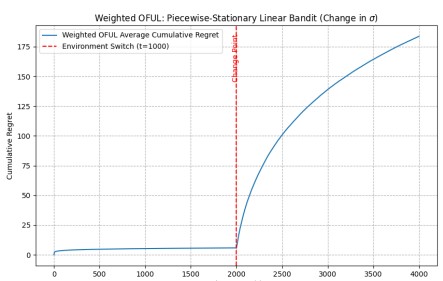

Figure 2: Weighted OFUL with Piecewise Assignment

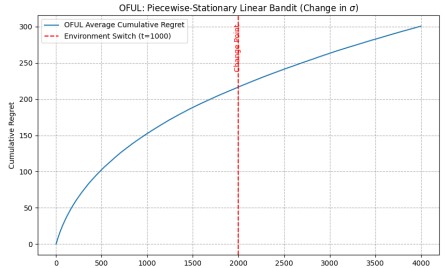

Figure 3: OFUL with Alternating Assignment

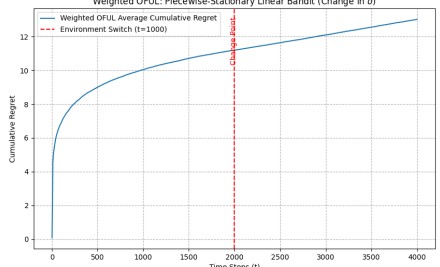

Figure 4: Weighted OFUL with Alternating Assignment

## B PROOF OF THEOREM 5.2

In this section, we prove the variance-dependent lower bound for adaptive variance sequences established in Theorem 5.2. We begin with the instance construction from Lemma 4.3 and establish the following constant-probability lower bound for the regret:

**Lemma B.1.** For a fixed variance threshold $\sigma$, number of rounds $K \geq 1.5d^2$, and any bandit algorithm Alg, for the instance constructed in Lemma 4.3, with probability at least $\Omega\big(1/\log(dK)\big)$, the regret is lower bounded by

$$\text{Regret}(K) \geq \frac{d\sqrt{K\sigma^2}}{16\sqrt{6}}.$$

Based on the constant-probability lower bound, we boost this probability by creating $L = \Omega\big(\log^2(dK)\big)$ independent instances with dimension $d' = d/L$ and number of rounds $K' = K/L$, where each instance follows the structure in Lemma 4.3 with i.i.d. sampled weight vectors. Under this construction, the total dimension of all instances is $d$, which can be represented as a $d$-

dimensional linear contextual bandit through orthogonal embedding, similar to our previous construction: for instance $i$, we augment its actions by padding zeros in dimensions reserved for other instances, ensuring actions from different instances only interact with their corresponding parameters. Here, we consider the case where the learner visits the instances in a cyclic manner and establish the following high-probability regret lower bound for the constructed instance:

**Lemma B.2.** For a fixed variance threshold $\sigma$, number of rounds $K \geq 1.5d^2$, and any bandit algorithm Alg, with probability at least $\Omega\big(1/\log(dK)\big)$, the regret is lower bounded by

$$\text{Regret}(K) \geq \Omega\big(d\sqrt{K\sigma^2}/\log^3(dK)\big).$$

With the help of this high-probability lower regret bound from Lemma B.2, we begin the proof of Theorem 5.2. Following a similar framework to the fixed-variance case, we first divide the rounds into groups based on their variance magnitude. Specifically, for any variance sequence $\{\sigma_1, \ldots, \sigma_K\}$, we partition the rounds into $L = \lceil \log_2 K \rceil + 1$ groups as follows:

$$\mathcal{K}_0 = \{k : \sigma_k \leq 1/K\},$$
$$\mathcal{K}_i = \{k : 2^{i-1}/K < \sigma_k \leq 2^i/K\}, \quad \text{for } i = 1, \ldots, L-1.$$

To address the unknown number of rounds $K_i = |\mathcal{K}_i|$, instead of constructing a single instance $\mathcal{M}_i$ for each group, we create $L$ instances $\mathcal{M}_{i,j}$, where $L = \lceil \log_2 K \rceil + 1$. Each instance $\mathcal{M}_{i,j}$ is constructed according to Lemma B.2 with dimension $d' = d/L^2$, variance $\sigma(i) = 2^{i-1}/K$ and number of rounds $K' = 2^{j-1}$. For each round $k$ in group $\mathcal{K}_i$, the learner receives a decision set $\mathcal{D}_i$ from one of the instances in $\{\mathcal{M}_{i,1}, \ldots, \mathcal{M}_{i,L}\}$ in a cyclic manner.

*Proof of Theorem 5.2.* According to Lemma B.2, for each instance $\mathcal{M}_{i,j}$, with probability at least $1 - 1/K^3$, the regret in the first $2^{j-1}$ visits is lower bounded by

$$\text{Regret}(2^{j-1}, \mathcal{M}_{i,j}) \geq \mathbb{I}(2^{j-1} \geq 1.5d'^2) \cdot \Omega\big(d'\sqrt{2^{j-1}\sigma^2(i)}/\log^3(d'K')\big), \qquad (B.1)$$

where the indicator reflects the requirement that $K' = 2^{j-1} \geq 1.5d'^2$. For simplicity, we define $\mathcal{E}$ as the event that (B.1) holds for all instances $\mathcal{M}_{i,j}$. By union bound, we have $\mathbb{P}(\mathcal{E}) \geq 1 - 1/K$. Conditioned on event $\mathcal{E}$, for an adaptive sequence and each corresponding group $\mathcal{K}_i$, due to the cyclic visiting pattern, each instance $\mathcal{M}_{i,j}$ is visited $|\mathcal{K}_i|/L$ times. There exists an instance $\mathcal{M}_{i,j}$ with matching interval for the round number, i.e., $2^{j-1} \leq |\mathcal{K}_i|/L \leq 2^j$. Therefore, we have

$$\sum_{k \in \mathcal{K}_i} \max_{\mathbf{x} \in \mathcal{D}_k} \langle \boldsymbol{\mu}_i, \mathbf{x} \rangle - \langle \boldsymbol{\mu}_i, \mathbf{x}_k \rangle$$
$$\geq \text{Regret}(2^{j-1}, \mathcal{M}_{i,j})$$
$$\geq \mathbb{I}(2^{j-1} \geq 1.5d'^2) \cdot \Omega\big(d\sqrt{2^{j-1}\sigma^2(i)}/\log^3(d'K')\big)$$
$$\geq \mathbb{I}(K_i \geq 3d'^2 L) \cdot \Omega\big(d\sqrt{K_i\sigma^2(i)}/\log^4(dK)\big)$$
$$\geq \Omega\Big(d'\sqrt{K_i\sigma^2(i)}/\log^3(dK) - d'\sqrt{3d'^2L\sigma^2(i)}/\log^4(dK)\Big)$$
$$\geq \Omega\bigg(d'\sqrt{\sum_{k \in \mathcal{K}_i} \sigma_k^2}/\log^4(dK) - \sqrt{3L}d'^2 \cdot \sigma(i)/\log^4(dK)\bigg), \qquad (B.2)$$

where the first inequality follows from $2^{j-1} \leq |\mathcal{K}_i|/L \leq 2^j$, the second inequality holds by the definition of event $\mathcal{E}$, the third inequality follows from $2^{j-1} \leq |\mathcal{K}_i|/L \leq 2^j$, the fourth inequality holds due to $\mathbb{I}(x \geq y)\sqrt{x} \geq \sqrt{x} - \sqrt{y}$, and the last inequality follows from the definition of group $\mathcal{K}_i$.

Taking a summation of (B.2) over all groups, the total regret can be lower bounded as follows:

$$\text{Regret}(K)$$
$$= \sum_{i=0}^{L-1} \sum_{k \in \mathcal{K}_i} \max_{\mathbf{x} \in \mathcal{D}_k} \langle \boldsymbol{\mu}_i, \mathbf{x} \rangle - \langle \boldsymbol{\mu}_i, \mathbf{x}_k \rangle$$
$$\geq \sum_{i=1}^{L-1} \Omega\bigg(d'\sqrt{\sum_{k \in \mathcal{K}_i} \sigma_k^2}/\log^4(dK) - \sqrt{3L}d'^2 \cdot \sigma(i)/\log^4(dK)\bigg)$$

$$\geq \Omega\left(\sum_{i=1}^{L-1} d/L^2 \cdot \sqrt{\sum_{k\in\mathcal{K}_i} \sigma_k^2 / \log^4(dK)} - 2\sqrt{3L}d^2/(L^4\log^4(dK))\right)$$

$$\geq \Omega\left(d/L^2 \cdot \sqrt{\sum_{i=1}^{L-1}\sum_{k\in\mathcal{K}_i} \sigma_k^2 / \log^4(dK)} - 2\sqrt{3L}d^2/(L^4\log^4(dK))\right), \tag{B.3}$$

where the first inequality follows from (B.2), the second inequality follows from the definition of variance threshold $\sigma(i)$ and dimension $d' = d/L^2$, and the last inequality holds due to $\sum_i \sqrt{x_i} \geq \sqrt{\sum_i x_i}$. In addition, for the group $\mathcal{K}_0$, we have

$$\sum_{k\in\mathcal{K}_0} \sigma_k^2 \leq \sum_{k\in\mathcal{K}_0} 1/K \leq 1, \tag{B.4}$$

where the first inequality follows from the definition of group $\mathcal{K}_0$ and the second inequality follows from $|\mathcal{K}_0| \leq K$. Therefore, we have

$$\text{Regret}(K)$$

$$\geq \Omega\left(d/L^2 \cdot \sqrt{\sum_{i=1}^{L-1}\sum_{k\in\mathcal{K}_i} \sigma_k^2 / \log^4(dK)} - 2\sqrt{3L}d^2/(L^4\log^4(dK))\right)$$

$$\geq \Omega\left(d/L^2 \cdot \sqrt{\sum_{i=1}^{L-1}\sum_{k\in\mathcal{K}_i} \sigma_k^2 - 1 / \log^4(dK)} - 2\sqrt{3L}d^2/(L^4\log^4(dK))\right)$$

$$\geq \Omega\left(d \cdot \sqrt{\sum_{i=1}^{L-1}\sum_{k\in\mathcal{K}_i} \sigma_k^2 / \log^6(dK)}\right),$$

where the first inequality follows from (B.3), the second inequality follows from (B.4), and the last inequality follows from the fact that $\sum_{k=1}^{K} \sigma_k^2 \geq \Omega(d^2)$. Thus, we complete the proof of Theorem 5.2. $\qquad\square$

## C  PROOF OF THEOREM 5.4

In this subsection, we provide the proof of Theorem 5.4. We begin by describing a simple algorithm:

1. The learner maintains an explored action set $\mathcal{A}$, which is initialized as empty.

2. For each decision set $\mathcal{D}_k$ in round $k$, if there exists an action $\mathbf{x}_k$ not in the spanning space of the explored action set $\mathcal{A}$, the learner:
   - Selects an action $\mathbf{x}_k$ and receives reward $r_k$;
   - Updates the explored set: $\mathcal{A} = \mathcal{A} \cup \{(\mathbf{x}_k, r_k)\}$.

3. Otherwise, when all actions lie in the spanning space of $\mathcal{A}$, the learner:
   - Estimates the reward for each action through linear combinations of $(\mathbf{x}, r) \in \mathcal{A}$;
   - Selects the action with maximum estimated reward.

It is worth noting that this algorithm assumes the received rewards $r_k$ have no noise to provide accurate estimates in step 3. While this assumption does not hold in general, when an adversary can choose the variance $\sigma_k$ based on the decision set $\mathcal{D}_k$, they can cooperate with the learner by setting:

- $\sigma_k = 0$ when step 2 is triggered (exploration);
- $\sigma_k = 1$ when step 3 is triggered (exploitation).

For a $d$-dimensional linear bandit problem, the explored action set satisfies $|\mathcal{A}| \leq d$. This implies the learner performs at most $d$ exploration steps with zero variance, while all remaining steps have variance one. Under this construction, the regret in the first $K$ rounds is upper bounded by:

$$\text{Regret}_{\text{Alg}}(K) \leq d,$$

where the total variance $\sum_{k=1}^{K} \sigma_k^2 = K - d \geq K/2$ (since $K \geq 2d$). Thus, through this cooperation between the adversary and learner, the $\widetilde{\Omega}(d\sqrt{\sum_{k=1}^{K}\sigma_k^2})$ lower bound is broken, completing the proof of Theorem 5.4.

# D  PROOF OF KEY LEMMAS

## D.1  PROOF OF LEMMA 4.3

In this subsection, we provide the proof of Lemma 4.3. When the variance threshold $\sigma = 1$, our construction reduces to the standard lower bound instances for linear contextual bandits (Zhou et al., 2021). Specifically, when the number of rounds $K$ satisfying $K \geq 1.5 \cdot d^2$, Zhou et al. (2021) provided the following variance-independent lower bound for these hard instances:

**Lemma D.1** (Lemma C.8, Zhou et al. 2021). For any bandit algorithm Alg, if the weight vector $\boldsymbol{\mu} \in \{-\Delta, \Delta\}^d$ is drawn uniformly at random from $\{-\Delta, \Delta\}^d$, then the expected regret over $K$ rounds is lower bounded by:

$$\mathbb{E}_{\boldsymbol{\mu}}[\text{Regret}(K)] \geq \frac{d\sqrt{K}}{8\sqrt{6}}.$$

With the help of Lemma D.1, we start the proof of Lemma 4.3.

*Proof of Lemma 4.3.* For any algorithm Alg for linear contextual bandit with fixed variance threshold $\sigma$, we construct an auxiliary algorithm Alg1 to solve the standard linear contextual bandit problem:

- At the beginning of each round $k \in K$, Alg1 observes the decision set $\mathcal{D}_k$ and sends it to Alg;
- Alg selects action $a_k \in \mathcal{D}_k$ based on the historical observations and delivers it to Alg1;
- Alg1 performs the action $a_k$, receives the reward $r_k$ and sends the normalized reward $\sigma \cdot r_k$ to Alg.

Now, we consider the performance of auxiliary algorithm Alg1 for the standard linear contextual bandit problem. It is worth noticing that the reward/noise in bandit instances for algorithm Alg1 and algorithm Alg only differ by a scalar factor $\sigma$, therefore for each instance, we have

$$\mathbb{E}[\text{Regret}_{\text{Alg}}(K)] = \sigma \cdot \mathbb{E}[\text{Regret}_{\text{Alg1}}(K)]. \tag{D.1}$$

If we randomly select a weight parameter vector $\boldsymbol{\mu} \in \{-\Delta, \Delta\}^d$, then according to Lemma D.1, the regret for Alg is lower bounded by

$$\mathbb{E}_{\boldsymbol{\mu}}[\text{Regret}_{\text{Alg}}(K)] = \sigma \cdot \mathbb{E}_{\boldsymbol{\mu}}[\text{Regret}_{\text{Alg1}}(K)] \geq \sigma \cdot \frac{d\sqrt{K}}{8\sqrt{6}} = \frac{d\sqrt{K\sigma^2}}{8\sqrt{6}},$$

where the equation holds due to (D.1) and the inequality holds due to Lemma D.1. Thus, we complete the proof of Lemma 4.3. $\qquad \square$

## D.2  PROOF OF LEMMA B.1

In this subsection, we provide the proof of Lemma B.1. We begin by recalling the OFUL algorithm in Abbasi-Yadkori et al. (2011) and its corresponding upper bound for the regret:

**Lemma D.2** (Theorem 3 in Abbasi-Yadkori et al. 2011). For any linear contextual bandit problem, with probability at least $1 - \delta$, the regret for OFUL algorithm in the first $K$ rounds is upper bounded by $\text{Regret}(K) \leq \widetilde{O}\big(d\sqrt{K \log(dK/\delta)}\big)$.

It is worth noting that the reward/noise in the instance construction from Lemma 4.3 only differs by a scalar factor $\sigma$ from the standard bandit. Therefore, as discussed in Section D.1, the regret in these two cases also only differs by a scalar factor $\sigma$. This leads to the following corollary:

**Corollary D.3.** For the instance construction from Lemma 4.3, there exists a constant $C$ such that with probability at least $1 - \delta$, the regret for OFUL algorithm in the first $K$ rounds is upper bounded by $\text{Regret}(K) \leq Cd\sqrt{K\sigma^2 \log(dK/\delta)}$.

With the help of Corollary D.3, we can begin the proof of Lemma B.1.

*Proof of Lemma B.1.* For any algorithm Alg, we construct an auxiliary algorithm Alg1 as follows:

- At the beginning of each round $k \in [K]$, Alg1 observes the decision set $\mathcal{D}_k$ and sends it to Alg;

- Alg selects action $a_k \in \mathcal{D}_k$ based on the historical observations and delivers it to Alg1;
- Alg1 performs the action $a_k$ and receives the reward $r_k$;
- Alg1 calculates the pseudo regret as:

$$\text{Regret}'(k) = \sum_{i=1}^{k} \frac{1}{3} + \frac{d}{\sqrt{96K}} - r_k.$$

If the pseudo regret is larger than $d\sqrt{K\sigma^2}/(8\sqrt{6}) + \sigma\sqrt{2K\log(2K/\delta)}$, Alg1 removes all previous information and performs the OFUL algorithm in all future rounds.

Based on the construction of the instances, whatever the weight vector $\boldsymbol{\mu}$ is, the optimal action is to select an action in the same direction as the weight vector, obtaining an expected reward of $1/3 + d/\sqrt{96K}$. Under this scenario, with probability at least $1 - \delta$, for any round $k \in [K]$, the difference between pseudo regret $\text{Regret}'(k)$ and true regret $\text{Regret}(k)$ can be upper bounded by

$$\left| \text{Regret}(k) - \text{Regret}'(k) \right| = \Big| \sum_{i=1}^{k} \epsilon_i \Big| \leq \sigma\sqrt{2K\log(2K/\delta)}, \tag{D.2}$$

where the inequality holds due to Lemma E.1 with the fact that the noise satisfies $\mathbb{E}[\epsilon_k|a_{1:k}, r_{1:k-1}] = 0$ and $|\epsilon_k| \leq \sigma$. Thus, according to the criterion of auxiliary algorithm Alg1, with probability at least $1 - \delta$, the regret of Alg1 before transitioning to OFUL is up to $d\sqrt{K\sigma^2}/(8\sqrt{6}) + 2\sigma\sqrt{2K\log(2K/\delta)}$. On the other hand, for the stage after transitioning to OFUL, Corollary D.3 suggests that with probability at least $1 - \delta$, the regret is no more than $Cd\sqrt{K\sigma^2\log(dK/\delta)}$. Therefore, with a selection of $\delta = 1/K$, we have

$$\mathbb{P}\big[\text{Regret}_{\text{Alg}_1}(K) \geq Cd\sqrt{K\sigma^2\log(dK^2)} + d\sqrt{K\sigma^2}/(8\sqrt{6}) + 2\sigma\sqrt{2K\log(2K^2)}\big] \leq 2/K. \tag{D.3}$$

For simplicity, let $R = Cd\sqrt{K\sigma^2\log(dK^2)} + d\sqrt{K\sigma^2}/(8\sqrt{6}) + 2\sigma\sqrt{2K\log(2K^2)}$ and we have

$$\mathbb{E}_{\boldsymbol{\mu}}[\text{Regret}_{\text{Alg}_1}(K)]$$
$$\leq \mathbb{P}\big[\text{Regret}_{\text{Alg}_1}(K) \geq R\big] \cdot K\sigma + \mathbb{P}\big[\text{Regret}_{\text{Alg}_1}(K) \geq d\sqrt{K\sigma^2}/(16\sqrt{6})\big] \cdot R$$
$$\quad + \mathbb{P}\big[\text{Regret}_{\text{Alg}_1}(K) \geq 0\big] \cdot d\sqrt{K\sigma^2}/(16\sqrt{6})$$
$$\leq 2\sigma + \mathbb{P}\big[\text{Regret}_{\text{Alg}_1}(K) \geq d\sqrt{K\sigma^2}/(16\sqrt{6})\big] \cdot \widetilde{O}(d\sqrt{K\sigma^2\log(dK)}) + d\sqrt{K\sigma^2}/(16\sqrt{6}),$$

where the first inequality holds due to $\mathbb{E}[X] \leq \mathbb{P}(X \geq x_1) \cdot R + \mathbb{P}(X \geq x_2) \cdot x_1 + \mathbb{P}(X \geq 0) \cdot x_2$ for $0 \leq X \leq R$ and $x_1 > x_2 > 0$, and the second inequality holds due to (D.3). Combining this result with the lower bound of expected regret in Lemma 4.1, we have

$$d\sqrt{K\sigma^2}/(8\sqrt{6}) \geq 2\sigma + \mathbb{P}\big[\text{Regret}_{\text{Alg}_1}(K) \geq d\sqrt{K\sigma^2}/(16\sqrt{6})\big] \cdot \widetilde{O}(d\sqrt{K\sigma^2\log(dK)})$$
$$\quad + d\sqrt{K\sigma^2}/(16\sqrt{6}),$$

which implies that

$$\mathbb{P}\big[\text{Regret}_{\text{Alg}_1}(K) \geq d\sqrt{K\sigma^2}/(16\sqrt{6})\big] \geq \Omega(1/\log(dK)). \tag{D.4}$$

In addition, according to the criterion of auxiliary algorithm Alg1 with (D.2), with probability at least $1 - \delta = 1 - 1/K$, Alg1 will not switch to the OFUL algorithm until the cumulative regret is larger than $d\sqrt{K\sigma^2}/(8\sqrt{6})$, which implies that

$$\mathbb{P}\big[\text{Regret}_{\text{Alg}}(K) \geq d\sqrt{K\sigma^2}/(16\sqrt{6})\big] \geq \mathbb{P}\big[\text{Regret}_{\text{Alg}_1}(K) \geq d\sqrt{K\sigma^2}/(16\sqrt{6})\big] - 1/K$$
$$= \Omega(1/\log(dK)).$$

Thus, we complete the proof of Lemma B.1. $\square$

### D.3 PROOF OF LEMMA B.2

In this subsection, we provide the proof of Lemma B.2.

*Proof of Lemma B.2.* Since the learner visits the instances in a cyclic manner, over all $K$ rounds, each instance $\mathcal{M}_i$ $(i = 1, 2, \ldots, L)$ is visited $K' = K/L$ times. As actions from different instances only interact with their corresponding parameters, according to Lemma B.1, for each instance $\mathcal{M}_i$, with probability at least $\Omega\big(1/\log(dK)\big)$, the regret is lower bounded by

$$\mathrm{Regret}(K', \mathcal{M}_i) \geq \frac{d'\sqrt{K'\sigma^2}}{16\sqrt{6}} = \frac{d\sqrt{K\sigma^2}}{16\sqrt{6} \cdot L^{1.5}}.$$

Note that the weight vectors for each instance are independently sampled, hence the probability that at least one instance has regret no less than $d\sqrt{K\sigma^2}/16\sqrt{6} \cdot L^{1.5}$ is at least

$$1 - \left(1 - \Omega\big(1/\log(dK)\big)\right)^L \geq 1 - 1/K^3.$$

Under this condition, the total regret can be lower bounded as:

$$\mathrm{Regret}(K) = \sum_{i=1}^{L} \mathrm{Regret}(K', \mathcal{M}_i) \geq \frac{d\sqrt{K\sigma^2}}{16\sqrt{6} \cdot L^{0.5}}. \tag{D.5}$$

Thus, we obtain a high-probability lower bound and complete the proof of Lemma B.2. $\qquad\square$

## E AUXILIARY LEMMAS

**Lemma E.1** (Azuma–Hoeffding inequality, Cesa-Bianchi & Lugosi 2006)**.** Let $\{\eta_k\}_{k=1}^{K}$ be a martingale difference sequence with respect to a filtration $\{\mathcal{G}_k\}$ satisfying $|\eta_k| \leq R$ for some constant $R$, $\eta_k$ is $\mathcal{G}_{k+1}$-measurable, $\mathbb{E}\big[\eta_k|\mathcal{G}_k\big] = 0$. Then for any $0 < \delta < 1$, with high probability at least $1 - \delta$, we have

$$\sum_{k=1}^{K} \eta_k \leq R\sqrt{2K\log(1/\delta)}.$$

