# OpenReview forum: "Variance-Dependent Regret Lower Bounds for Contextual Bandits"
_ICLR.cc/2026/Conference — ICLR 2026 Poster_

### Official Review · Reviewer_dB4e · 2025-10-27

**Soundness:** 3
**Presentation:** 3
**Contribution:** 3
**Rating:** 6
**Confidence:** 3

**Summary:**

The paper studies the variance-dependent lower bound for the linear contextual bandits.
While the previously known lower bound required the specific choice of variance-sequences $\{\sigma^2_k\}$ and only considered the total fixed variance budget,  this paper extends the analysis to more general variance sequences.

 For the prefixed sequence, where the variance sequence is revealed to the learner at the beginning of the learning process, the paper establishes an expected lower bound of　$\tilde{\Omega}( d \sqrt{\sum_{k=1}^K \sigma^2_k })$．


For the adaptive sequence: for the weak adversary (who must generate the $\sigma_k$ before observing the decision set $D_k$),
they provide  the high probability lower bound of $\tilde{\Omega}( d \sqrt{\sum_{k=1}^K \sigma^2_k })$.


For the strong adversary who can generate $\sigma_k$ after observing the decision set $\mathcal{D}_k$, they show that the counter algorithm can collaborate with the adversary and achieve $O(d)$ regret even when $\sum \sigma^2_k =\Omega(K)$. This is due to the construction where the learner can effectively observe noise-free rewards in certain rounds depending on $\mathcal{D}_k$ and action set, obtaining full exploration in just $d$ rounds. In the remaining rounds, even when the variance is one, the regret remains  $O(d)$ when $K >2d$.

**Strengths:**

- The heteroscedastic linear contextual bandit  is a general model and it is important to investigate the lower bound of the cumulative regret. The previous result  only considered the total fixed variance budget.  This paper extends the analysis to handle more general variance sequences.

- The lower bounds match the upper bound in Zhao et al. (2023) up to logarithmic factors.
- A peeling technique for prefixed variance sequence and the orthogonal construction of decision sets across different groups are novel techniques to provide novel hard-to-learn instances.
Theorem 5.2 is the first high-probability lower bound for linear contextual bandit.

**Weaknesses:**

- Theorem 5.2 has additional dependency of $1/\log^6(dK)$, which is logarithmic but still somewhat loose compared to the upper bound.
- See also the question below regarding the assumptions in Theorem 5.4.

**Questions:**

For the strong adversary who can generate $\sigma_k$ after observing the decision set $\mathcal{D}_k$, the reason why a counter algorithm cooperating with the adversary incurs only $O(d)$ regret is that the construction allows the adversary to assign zero variance in certain rounds. What if we restrict the strong adversary to generate strictly positive (non-zero) variance?

---

> ### Author Response · Authors · 2025-11-19
> **Reply to Review**
>
> --------
>
> Thank you for your positive and thoughtful feedback. We address your comments below:
>
>
> ----
>
> **Q1**: Theorem 5.2 has additional dependency of  $1/\log^6(dK)$, which is logarithmic but still somewhat loose compared to the upper bound.
>
>
> **A1**: Thanks for your comment. We will try to improve the logarithmic dependency in the future work.
>
> -----
>
>
> **Q2**: For the strong adversary who can generate $\sigma_k$ after observing the decision set $\mathcal{D}_k$, the reason why a counter algorithm cooperating with the adversary incurs only $O(d)$ regret is that the construction allows the adversary to assign zero variance in certain rounds. What if we restrict the strong adversary to generate strictly positive (non-zero) variance?
>
>
> **A2**:For strictly positive variance, the adversary may not guarantee a constant regret like in Theorem 5.4 or Corollary 5.7, but it will still significantly impact the construction of the lower bound.
>
> The main motivation is based on the observation that high variance has a high impact on exploration—making it hard for the learner to collect reliable information—but has a low impact on exploitation once the optimal action is well-identified. Therefore, the adversary can allocate a lower variance level ($\sigma_k^2$) in the exploration stage and allocate a higher variance level ($\sigma_k^2$) in the exploitation stage to disrupt the lower bound construction. In the proof of Theorem 5.4, we selected the zero variance level ($\sigma_k^2=0$) in the exploration stage, but the motivation still holds for small positive variance.
>
>
> While a full analysis for a general adaptive decision set with a strictly positive variance floor ($\sigma_k \ge \sigma_{\min}$) is challenging, the core concept can be illustrated using the fixed decision set case (Corollary 5.7) as a counterexample.
>  Consider a scenario with variance sequence $\sigma_1, \dots, \sigma_{K/2} = \sigma_{\min}, \sigma_{K/2+1}, \dots, \sigma_{K}=1$. For the first period $[1, K/2]$, there is a uniform variance of $\sigma_{\min}$, and the SAVE algorithm can already provide an $\mathcal{O}(d\sqrt{\sigma_{\min}^2 K/2} + d)$ regret guarantee. For the second period $[K/2+1, K]$, we can trivially replay each action in the first period and still maintain regret of $\mathcal{O}(d\sqrt{\sigma_{\min}^2 K/2} + d)$.
>
> Even though the total cumulative variance over all $K$ rounds may be large ($\Omega(K)$), the final regret remains bounded by $\mathcal{O}(d\sqrt{\sigma_{\min}^2 K} + d)$. This explicitly shows that the strategic allocation of a small positive variance in the critical exploration stage prevents the lower bound from scaling with the large total variance, thus illustrating the fragility of the lower bound construction with strong adversary. We note that setting $\sigma_{\min}=1$ recovers the standard $\Omega(d\sqrt{K})$ regret bound for linear bandit, and $\sigma_{\min}=0$ recovers the $\Omega(d)$ regret bound in Corollary 5.7.
>
> -----

---

### Official Review · Reviewer_9a2R · 2025-10-28

**Soundness:** 3
**Presentation:** 3
**Contribution:** 3
**Rating:** 6
**Confidence:** 2

**Summary:**

The paper provides the first high probability lower bounds on variance dependent regret for linear contextual bandits. These bounds match the known upper bounds up to logarithmic factors under varied assumptions on the adversarial strength in deciding and revealing variance sequences. In proving these novel results, the authors devise a "peeling" proof machinery which has broader implications beyond the studied problem instance.

**Strengths:**

The derivation of variance-dependent lower bounds is an important and challenging problem in bandits literature. Achieving near-tightness (up to logarithmic terms) under a generalized heteroscedastic setting is a significant theoretical advance.

The proposed "peeling" or "grouping" technique appears to be novel and essential in obtaining these sharper bounds, suggesting its potential applicability to other structured problems (see my questions for more discussion on the implications of this machinery).

The analysis under varied assumptions on adversarial strength adds depth and robustness to the result.

**Weaknesses:**

The bibliography contains several critical errors that must be corrected before publication:
 - The first entry appears to be duplicated.
- The citation for Dani (2008) is reportedly incorrect and should instead refer to the CoLT (Contextual Linear Threshold) paper.
- Several cited preprints have since been formally published (journal articles, updated conference proceedings) and must be updated to their final, official citations.

The statement of Theorem 4.1 (and many such statements) are imprecise and somewhat confusing. The authors should rigorously re-examine this statement for technical correctness and clarity.

The paper introduces a notion of "collaboration" or a specific form of an adversary. Claiming "near-tight" bounds is only meaningful if the lower bound is tight against the existing upper bounds (like SAVE or similar) under the same set of assumptions. If the adversarial definition is fundamentally different from the standard setting used for upper bounds, the claim of near-tightness seems misleading? Feel free to clarify this point for me, I am unfamiliar with this literature.

The writing quality is generally acceptable, but the material can be opaque at times. Given that most detailed proofs are deferred to the appendix, the main body should dedicate more space to building intuitive understanding of the key proof ideas. Specifically, the high-level intuition behind the "peeling" technique needs to be expanded in the main text for the uninitiated reader.

**Questions:**

Proof Rigor (Lemma A.2): Prior to Lemma A.2, the argument mentions considering a specific cyclic instance where the learner visits instances in a cyclic manner. However, the resulting lower bound is claimed to hold for any bandit algorithm. Please explicitly include or reference the standard information-theoretic reduction that proves the bound derived from this specific instance generalizes to all valid bandit algorithms?

Heteroscedastic Variance Utility: Can you provide an expanded explanation of the utility of leveraging "heteroscedastic variance information". Furthermore, what key technical obstacles generally prevent obtaining variance-dependent lower bounds in bandit problems, and what was the unique aspect of the LCB problem that allowed the authors to overcome this limitation here?

Peeling Technique and Extensions: Please highlight the novelty and essential nature of this grouping/peeling proof technique. What is the fundamental mechanism that allows it to achieve tighter, variance-dependent results compared to prior methods? Can the authors speculate on how this technique might be useful for other problems or domains, particularly in terms of extending the ideas to more complex bandit style problems beyond the linear contextual case?

Broader Context: Please situate this result within the broader literature of lower bounds for bandit type problems (i.e., not just linear contextual). Does this result have broader implications based on the established hierarchy of these problems (e.g., standard MAB, stochastic contextual, LCB)?

Adversarial Bound (Collaboration): Regarding the "collaboration" notion, please clarify the specific regret bound on SAVE (or the best known algorithm in this setup) that your lower bound is being compared against under your definition of the adversary.

---

> ### Author Response · Authors · 2025-11-18
> **Reply to Review**
>
> ----
>
> Thank you for your positive and thoughtful feedback. Below we reply to your comment point-by-point.
>
> ----
>
> **Q1**: The bibliography contains several critical errors that must be corrected before publication
>
>
> **A1**: Thanks for your comment, we have revised the bibliography and uploaded the new version.
>
> -----
>
> **Q2**: The statement of Theorem 4.1 (and many such statements) are imprecise and somewhat confusing. The authors should rigorously re-examine this statement for technical correctness and clarity.
>
>
> **A2**:  We have rigorously re-examined the statement of Theorem 4.1 for technical correctness and clarity and uploaded the new version.
>
> The corrected statement is:"Let $d>1$ and consider any prefixed sequence of variances $\{\sigma_1,...,\sigma_K\}$ satisfying $\sum_{k=1}^K \sigma_k^2 \geq 1 + 384d^2$. For any algorithm $\mathrm{Alg}$, there exists a hard linear contextual bandit instance such that each action $a\in \mathcal{D}_k$ in round $k$ has variance bounded by $\sigma_k^2$. For this instance, the expected regret of algorithm $\mathrm{Alg}$ over $K$ rounds is lower bounded by:
>
> $        E[\mathrm{Regret}(K)\big]\ge {\Omega}\Big(d\sqrt{\textstyle \sum_{i=1}^K\sigma_{k}^2}/(\log K)\Big).$
>
>
> -----
>
>
> **Q3**: The paper introduces a notion of "collaboration" or a specific form of an adversary. Claiming "near-tight" bounds is only meaningful if the lower bound is tight against the existing upper bounds (like SAVE or similar) under the same set of assumptions. If the adversarial definition is fundamentally different from the standard setting used for upper bounds, the claim of near-tightness seems misleading? Feel free to clarify this point for me, I am unfamiliar with this literature.
>
> Adversarial Bound (Collaboration): Regarding the "collaboration" notion, please clarify the specific regret bound on SAVE (or the best known algorithm in this setup) that your lower bound is being compared against under your definition of the adversary.
>
> **A3**: Thanks for the comment, and we want to clarify that: In fact, the theoretical guarantee of the SAVE algorithm only requires that the variance at round k is bounded by $\sigma_k^2$, and it achieves a regret guarantee of $\tilde{O}\left(d \sqrt{\sum_{k=1}^K \sigma_k^2}\right)$ in all three different cases (where the variance is prefixed, or chosen before observing the decision, or after observing the decision set).
>
> In comparison, we have three distinct lower bounds that correspond to the three cases defined by the adversary's timing in selecting $\sigma_k$. For each scenario, we can directly compare our respective lower bound, with the same upper bound in the SAVE algorithm, $\tilde{O}\left(d \sqrt{\sum_{k=1}^K \sigma_k^2}\right)$.
>
> -----
>
>
> **Q4**: The writing quality is generally acceptable, but the material can be opaque at times. Given that most detailed proofs are deferred to the appendix, the main body should dedicate more space to building intuitive understanding of the key proof ideas. Specifically, the high-level intuition behind the "peeling" technique needs to be expanded in the main text for the uninitiated reader.
>
>
> **A4**:  Thank you for this constructive feedback. We agree that a clearer intuitive explanation of our proof technique is necessary. We have expanded the discussion in the main body, specifically by enhancing Section 4.2 (which covers the warm-up case with a prefixed variance sequence and the construction of the hard instance). We have uploaded the new version incorporating the following paragraph to illustrate the high-level intuition:
>
> “The basic idea for the lower regret bound is to assign different orthogonal sub-instances based on the range of the variance $\sigma_k$ at the beginning of each round. This method ensures that each orthogonal instance will be learned with comparable variance, which makes it easier to derive a tighter lower regret bound. Finally, since the orthogonal instances cannot provide mutual information, the total regret can be decomposed into the summation of the regret accumulated in each sub-instance.”
>
>
>
>
>
>
> -----

---

> > ### Author Response · Authors · 2025-11-18
> > **Reply to Review**
> >
> > -----
> >
> > **Q5**: Proof Rigor (Lemma A.2): Prior to Lemma A.2, the argument mentions considering a specific cyclic instance where the learner visits instances in a cyclic manner. However, the resulting lower bound is claimed to hold for any bandit algorithm. Please explicitly include or reference the standard information-theoretic reduction that proves the bound derived from this specific instance generalizes to all valid bandit algorithms?
> >
> >
> > **A5**: First, we want to clarify that for the linear contextual bandit problem, the decision set (the set of actions) is not chosen by the algorithm; rather, the algorithm receives a decision set and must select an action from it. Therefore, in the proof of Lemma A.2, the sequence of decision sets is a part of the lower bound construction, and this construction works for all bandit algorithms. In addition, as shown in Corollary 5.7, if the decision set were fixed (meaning we could not adaptively select the decision set sequence), the lower bound would not hold.
> >
> > Second, in the proof of Lemma A.2, we create multiple instances. The lower bound for each instance can be traced back to Lemma C.1 (or Lemma C.8 in Zhou et al. 2021), which is indeed based on information-theoretic analysis.
> >
> > ----
> >
> > **Q6**:
> > Heteroscedastic Variance Utility: Can you provide an expanded explanation of the utility of leveraging "heteroscedastic variance information". Furthermore, what key technical obstacles generally prevent obtaining variance-dependent lower bounds in bandit problems, and what was the unique aspect of the LCB problem that allowed the authors to overcome this limitation here?
> >
> > **A6**: First, we clarify the utility of leveraging heteroscedastic variance information: For a standard bandit algorithm, if a homogeneous variance upper bound exists (e.g., $\sigma_k \leq \sigma$ for all $k$), the regret bound would be $\tilde{O}(d\sqrt{\sigma^2 K})$, where $K$ is the number of rounds. However, in most cases, the variance is not fixed, and using a single homogeneous variance upper bound for all rounds may be loose in some rounds, which fails to capture the fundamental difficulty of the problem. In comparison, leveraging heteroscedastic variance information (i.e., treating $\sigma_k$ as a sequence of potentially different values) leads to a much tighter regret bound, $\tilde{O}\left(d \sqrt{\sum_{k} \sigma_k^2}\right)$.
> >
> > Second, the main difficulty in obtaining variance-dependent lower bounds comes from dealing with an adaptive variance sequence. Generally speaking, high variance has a high impact on exploration—making it hard for the learner to collect reliable information—but has a low impact on exploitation once the optimal action is well-identified. In this context, as shown in Corollary 5.7, having a small variance in the early stage and a later large variance can break down the lower bound for the standard stochastic linear bandit problem where the decision set is fixed.
> >
> > This is the key point where we rely on the Linear Contextual Bandit (LCB) setting: we can allocate different contextual sets to the learner based on the variance range (via the peeling technique). This ability allows us to construct a hard instance that avoids the early exploration stage having much smaller variance than the later exploitation stage, thus successfully circumventing the standard limitation.
> >
> > ----
> >
> > **Q7**:
> > Peeling Technique and Extensions: Please highlight the novelty and essential nature of this grouping/peeling proof technique. What is the fundamental mechanism that allows it to achieve tighter, variance-dependent results compared to prior methods? Can the authors speculate on how this technique might be useful for other problems or domains, particularly in terms of extending the ideas to more complex bandit style problems beyond the linear contextual case?
> >
> > **A7**:
> > The novelty and essential nature of our peeling proof technique are twofold:
> >
> > Lower Bound Application: While the peeling method is common for deriving upper bounds in algorithm design, its application has been strictly limited to that domain. Our work is the first to successfully introduce and apply this method to establish lower bounds. This results in a tight, variance-aware bound that holds over a general, adaptive variance sequence.
> >
> > High-Probability Bounds: To the best of our knowledge, our result provides the first high-probability lower bound for linear contextual bandits, moving beyond the standard expected lower bounds found in prior literature. This technique is of independent interest and can be used to establish high-probability lower bounds for a wider array of settings. We have added a dedicated discussion on this topic in Remark 5.3.
> >
> >
> >
> >
> >
> > ----

---

> ### Author Response · Authors · 2025-11-18
> **Reply to Review**
>
> ----
> **Q8**: Broader Context: Please situate this result within the broader literature of lower bounds for bandit type problems (i.e., not just linear contextual). Does this result have broader implications based on the established hierarchy of these problems (e.g., standard MAB, stochastic contextual, LCB)?
>
> **A8**:  Thank you for the comment. We would like to discuss some related literature for different settings.
>
> For the multi-armed bandit (MAB) problem, previous works [1] have indeed provided variance-dependent upper bounds on the cumulative regret, typically expressed as $\mathcal{O}\left(\sum_{a}\frac{\sigma_a^2}{\Delta_a}\right)$, where $\Delta_a$ is the sub-optimality gap of arm $a$ against the optimal arm, and $\sigma_a^2$ is the variance for arm $a$. It is important to note that this MAB result relies fundamentally on the assumption that the arms are fixed. Consequently, the sub-optimality gap ($\Delta_a$) and the variance ($\sigma_a^2$) are assumed to remain constant across all rounds. In comparison, our focus is on the linear contextual bandit (LCB) problem, where the decision set ($\mathcal{D}_k$) changes adaptively and is not comparable with the regret bound in the MAB problem.
>
> For the stochastic linear bandit problem, where the decision set is fixed, having a small variance in the early stage and a later large variance can impact performance. As shown in Corollary 5.7, the high variance has a high impact on exploration—making it hard for the learner to collect reliable information—but has a low impact on exploitation once the optimal action is well-identified. For the stochastic contextual bandit, where each contextual vector in the decision set $\mathcal{D}_k$ is i.i.d. selected, the situation is closer to the stochastic linear bandit. In this case, exploration will focus on the early stage and the later stage will focus on exploitation, which may result in a similar constant regret as when the decision set is fixed.
>
> However, as clarified in A6, our method relies on adaptively allocating a different decision set to the agent to prevent the early exploration stage from having a much smaller variance than the later exploitation stage. Our method can be extended to broader bandit settings that allow an adaptive decision set, such as contextual generalized bandits or contextual bandits with general function approximation.
>
>
> [1]  Exploration-exploitation trade-off using variance estimates in multi-armed bandits
>
> We have added a new paragraph discussing variance-dependent regret bounds for MAB in the related work section of the revised paper.
>
> -----

---

### Official Review · Reviewer_uLK1 · 2025-10-31

**Soundness:** 3
**Presentation:** 3
**Contribution:** 2
**Rating:** 6
**Confidence:** 3

**Summary:**

The paper studies linear contextual bandits with round-varying noise and asks whether the variance-aware upper bounds $\tilde{O}(d\sqrt{\sum_t \sigma_t^2})$ are actually tight. It shows that when the whole variance sequence is fixed in advance, for any sequence one can build a contextual instance on which every algorithm incurs $\tilde{\Omega}(d\sqrt{\sum_t \sigma_t^2})$ regret matching known upper bounds. For an adaptive but weak adversary it further gets a high-probability lower bound with more high-order polylog factors. Finally, it shows that for a strong adaptive adversary, such variance-dependent lower bounds are impossible, giving a comprehensive understanding of second-order bound for linear contextual bandit.

**Strengths:**

1. This paper closes a theoretical gap by deriving a new variance-dependent lower bound that matches previous upper bounds. This improves previous lower bound that depends on total-variance budget and suboptimal $d$ factor for linear contextual bandit.

2. This paper is technically neat, the “variance-level peeling + orthogonal subproblems” construction is clean and reusable; the high-probability version for adaptive variance is nontrivial.

3. The paper gives a comprehensive discussion for the separation of weak vs. strong adaptive adversaries and shows why the strong adversary cannot admit such lower bounds.

**Weaknesses:**

I did not see major weakness. The current lower bound relies on being able to assign (almost) orthogonal decision sets to different variance groups. That’s fine for an information-theoretic contextual lower bound with adversarially chosen context, but less reflective of i.i.d. contexts. Is it possible to improve the lower bound for linear contextual bandit with stochastic context?

**Questions:**

See weakness.

---

> ### Author Response · Authors · 2025-11-17
> **Reply to Review**
>
> ---
>
> Thank you for your positive and thoughtful feedback. We address your comment below:
>
> ---
>
> **Q1**: Is it possible to improve the lower bound for linear contextual bandit with stochastic context?
>
>
> **A1**:  We have checked the construction of our hard-to-learn instance and it does not work for the stochastic context. In detail, our instance relies on creating several different sub-instances with orthogonal decision sets and assigning them to the learner based on the range of the variance. For i.i.d. contexts, it is impossible to perform such allocation.
>
> In addition, the key intuition from the lower bound stems from the fact that high variance ($\sigma_k^2$) will have a high impact on exploration—making it hard for the learner to collect reliable information—but will have a low impact on exploitation once the optimal action is well-identified. That is why we need a carefully assigned sub-instances based on the range of the variance.
>
> However, for an i.i.d. contexts set, the exploration will focus on the early stage and the later stage will focus on exploitation. This is close to the situation where the decision set is fixed (in Corollary 5.7). For a prefixed sequence with $\sigma_1, \ldots, \sigma_{\Omega(d)}=0$, and $\sigma_{\Omega(d)+1}, \ldots, \sigma_{N}=1$, an algorithm can identify the latent vector in the first $\Omega(d)$ exploration rounds, and later the variance $\sigma_k=1$ will not impact the performance in the exploitation stage. Therefore, the i.i.d. contexts set may have a similar constant regret as the decision set is fixed (in Corollary 5.7), even the total variance is linear with respect to the number of rounds $K$.
>
> ---

---

### Official Review · Reviewer_vE2i · 2025-10-31

**Soundness:** 3
**Presentation:** 2
**Contribution:** 2
**Rating:** 2
**Confidence:** 3

**Summary:**

This work studies linear contextual bandits with changing decision sets and aims to characterize optimal variance-based regret rates, depending on the sequence of reward variances. Previous lower bounds were shown in terms of a fixed total variance budget, and had a mismatch in the dimension dependence. This paper strengthens the lower bound in these regards for various modeling assumptions on the adversary.

**Strengths:**

* The paper refines known lower bounds in linear contextual bandits in terms of dimension dependence and generalized variance sequences, which requires novelties in the analysis such as a refined peelilng technique and separate analyses of different variance values.

**Weaknesses:**

* The presentation seems lacking in motivation. am not sure what the real world applicability of the various adversary (prefixed vs. weak vs. strong) scenarios are and there could also be more discussion of whether such refined variance-sequence based lower or upper regret bounds are known for the simpler finite-armed bandit case.
* There are also no experiments empirically demonstrating the regret bounds exhibit such tight dependences.
* I also find the presentation a bit confusing or lacking rigor in that, at times, especially Theorem 5.4. The writing at times (e.g., Remark 3.1) suggests the adversary may cooperate with the learner, but I'm not sure how to interpret this as "adversary" in bandits typically represents a worst-case environment or environment-generating process. I think this may be due to lack of clarity on the order of quantifiers in theorem statements. Perhaps a diagram or table would make it easier to understand. The utility of Theorem 5.4 is unclear as statements of the kind "there exists an adversary" implies such that there is an algorithm that can get small regret seem unsurprising. It is more useful to understand what is the worst-case behavior here, or is the strong adversary somehow weaker than the prefixed one?
* Because of the wording of Theorem 5.4, it's unclear whether the statement "it's impossible to derive a variance-dependent lower bound if the adversary can determine the variance after observing the decision set" really follows as Theorem 5.4 concerns a _specific_ adversary and not the worst-case one.
* Furthermore, the problem is quite complex as the decision sets are also being generated adversarailly (as it adaptive or oblivious?) and the adversary may decide the variance before or after the decision set which seems to affect the bounds. In particular, some intuition on why the weak and strong adversary leads to different rates would be helpful.
* The scope of technical novelties of this work also seem limited, and it's a bit unsurprising that general variance sequence-based bounds can be derived using a grouping/peeling argment. Can the authors elaborate on difficulties of the general function approximation setting or discuss wider applicability of techniques introduced here?

**Questions:**

Please see weaknesses above.

---

> ### Author Response · Authors · 2025-11-17
> **Reply to Review**
>
> -----
>
> Thank you for your thoughtful feedback. Below we address your concerns point-by-point.
>
> ------
>
> **Q1**: The presentation seems lacking in motivation. I'm not sure what the real-world applicability of the various adversary (prefixed vs. weak vs. strong) scenarios are.
>
> **A1**:
>
> Our work is focused on the theoretical side of bandit algorithms.The motivation of studying various adversaries stems directly from the differences in existing literature that proves different regret upper bounds for linear contextual bandits, where the decision set is chosen in different ways. We summarize these settings as follows:
>
> 1. In the oblivious case, where all decision sets $\mathcal{D}_k$ are predefined at the beginning, algorithms like SupLinUCB [1] achieve a regret guarantee of $\mathcal{O}(\sqrt{dK})$.
>
> 2. In the adaptive case, where the decision set $\mathcal{D}_k$ can depend on previous actions and rewards, standard algorithms like OFUL [2] can only guarantee a worse rate of $\mathcal{O}(d\sqrt{K})$.
>
> In [1,2], the environment is typically assumed to have a simple noise model with the same variance across rounds (e.g., $\sigma_1^2, \ldots, \sigma_K^2 = R^2$). Later,  [3,4] generalized this to a general variance sequence, and provided variance-dependent upper regret bounds for the adaptive variance setting.
>
> Following this established line of inquiry, the environment's ability to adapt its variance level impacts the fundamental difficulty and thus the theoretical lower bound. Therefore, studying the different variance sequence scenarios is well-motivated:
>
> 1. Prefixed vs. Adaptive Variance: It is necessary to consider the difference in fundamental difficulty between a prefixed variance sequence and an adaptive sequence.
>
> 2. Adaptive Interactions (Weak vs. Strong): When both the decision set $\mathcal{D}_k$ and the variance level $\sigma_k^2$ are adaptive, we must characterize the impact of the timing and order of decisions. The strong adversary represents the case where the decision set is chosen first, while the weak adversary represents the sequence where the variance level is chosen first. These scenarios represent different levels of information asymmetry available to the environment-generating process, which is critical for establishing tight information-theoretic lower bounds.
>
> [1] Contextual Bandits with Linear Payoff Functions
>
> [2] Improved Algorithms for Linear Stochastic Bandits
>
> [3] Nearly minimax optimal reinforcement learning for linear mixture markov decision processes.
>
> [4] Variance-dependent regret bounds for linear bandits and reinforcement learning: Adaptivity and computational efficiency.
>
> ----
> **Q2**: There could also be more discussion of whether such refined variance-sequence based lower or upper regret bounds are known for the simpler finite-armed bandit case.
>
> **A2**: For the multi-armed bandit problem, previous works [5] have indeed provided variance-dependent upper bounds on the cumulative regret, typically expressed as $\mathcal{O}\left(\sum_{a}\frac{\sigma_a^2}{\Delta_a}\right)$, where $\Delta_a$ is the sub-optimality gap of arm $a$ against the optimal arm, and $\sigma_a^2$ is the variance for arm $a$.
>
> It is important to note that this MAB result relies fundamentally on the assumption that the arms are fixed. Consequently, the sub-optimality gap ($\Delta_a$) and the variance ($\sigma_a^2$) are assumed to remain constant across all rounds.
>
> In sharp contrast, our focus is on the linear contextual bandit problem where the decision set ($\mathcal{D}_k$) changes adaptively. This change depends on the history of actions and rewards, meaning the set of available arms (and even the size of the action set) is not fixed.
>
> We have added a new paragraph discussing variance-dependent regret bounds for MAB in the related work section of the revised paper.
>
> [5] Exploration-exploitation trade-off using variance estimates in multi-armed bandits
>
>
> ----
>
> **Q3**: There are also no experiments empirically demonstrating the regret bounds exhibit such tight dependencies.
>
> **A3**: Our work focuses on establishing theoretical lower bounds on the regret achievable by any bandit algorithm. It is impossible to show the performance for all bandit algorithms empirically, as the lower bound represents an information-theoretic limit across a class of problems, not performance on a single algorithm.
>
> We are currently running simulation experiments using the specific hard instance constructed in Theorem 4.1. We are testing two representative algorithms: SAVE and OFUL. We plan to include these results in the revised paper to empirically verify that their performance matches the dependencies defined by our theoretical lower bound.
>
> ----

---

> ### Author Response · Authors · 2025-11-17
> **Reply to Review**
>
> **Q4**: I also find the presentation a bit confusing or lacking rigor in that, at times, especially Theorem 5.4. The writing at times (e.g., Remark 3.1) suggests the adversary may cooperate with the learner, but I'm not sure how to interpret this as "adversary" in bandits typically represents a worst-case environment or environment-generating process. I think this may be due to lack of clarity on the order of quantifiers in theorem statements. Perhaps a diagram or table would make it easier to understand.
>
> **A4**:  Sorry for the confusion, we must clarify that the definition of adversary in the context of our lower bound construction differs from its use in upper bound analysis, particularly when discussing adaptation.
>
> ### **Adversary in Upper vs. Lower Bound**
>
> 1.  Upper Bound Analysis: The adversary here represents the worst-case environment that selects the sequence of decision sets ($\mathcal{D}_k$). Prior work shows that even if this "adversary" chooses the decision set adaptively (based on history), the algorithm can still achieve an efficient regret guarantee. This proves the algorithm's robustness against the hardest possible input sequence.
> 2.  Lower Bound Construction (Our Focus): The adversary is the entity we construct to define the information-theoretic limit ($\sup_{\mathcal{I}}$). We aim to establish a theoretical lower bound for any adaptive variance sequence ($\sigma_k^2$), even when an adversary can choose $\sigma_k^2$ based on previous observations. In this case, the adversary's choices are aimed at breaking the information-theoretic construction of the lower bound—meaning they effectively collaborate with the learner to reduce the regret.
>
> ### **Clarification on Adaptive Timing**
>
> In our work, we explicitly use the word "adversary" to highlight that both the decision set $\mathcal{D}_k$ and the variance level $\sigma_k^2$ are adaptive, and we want to draw attention to the fundamental difficulty related to the ordering of these two adaptive decisions.
>
> When both $\mathcal{D}_k$ and $\sigma_k^2$ are adaptive in each round, the timing defines the information asymmetry:
>
>  1. Strong Adversary: The environment chooses the decision set $\mathcal{D}_k$ first. This represents a more challenging environment where the adversary can generate a strong adaptive variance sequence that depends on the chosen decision set $\mathcal{D}_k$.
>
>  2. Weak Adversary: The environment chooses the variance level $\sigma_k^2$ first. This represents a weaker adaptive variance sequence that cannot depend on the decision set $\mathcal{D}_k$ in that round.
>
> These scenarios help us precisely characterize the impact of this adaptive interplay on the resulting theoretical lower bound.
>
> ### **Revision**
> To improve clarity, we will revise the terminology in the paper from "Strong/Weak Adversary" to Strong/Weak Adaptive Variance Sequence.
>
>
>
> -----
>
> **Q5**:  The utility of Theorem 5.4 is unclear as statements of the kind "there exists an adversary" implies such that there is an algorithm that can get small regret seem unsurprising. It is more useful to understand what is the worst-case behavior here, or is the strong adversary somehow weaker than the prefixed one? Because of the wording of Theorem 5.4, it's unclear whether the statement "it's impossible to derive a variance-dependent lower bound if the adversary can determine the variance after observing the decision set" really follows as Theorem 5.4 concerns a specific adversary and not the worst-case one.
>
>
> **A5**:  As discussed in A4, our primary aim is to provide a lower bound that holds for any adaptive variance sequence. The terms Strong Adversary and Weak Adversary distinguish the level of adaptivity (strong/weak) of that variance sequence. The Strong Adversary leads to a large class of possible variance sequences, making it inherently harder to construct instances whose lower bound holds for all sequences in that large class.
>
> We will clarify the claims regarding the strong adaptive setting by making the following revision to Theorem 5.4:
>
> For any linear contextual bandit problem with any adaptive decision set sequence $(D_1, ... , D_K)$ and any number of rounds $K > 2d$, there exists a strong adaptive variance sequence such that there exists an algorithm whose regret in the first $K$ rounds is upper bounded by $\text{Regret}(K) \leq d$, where the total variance $\sum_{k=1}^{K} \sigma^2_k \geq K/2$.
>
> In Theorem 5.4, we show that there exists at least one strong adaptive variance sequence for which the achievable regret is constant, specifically $\mathcal{O}(d)$. This result demonstrates that the environment has too much flexibility, preventing the existence of a non-trivial variance-dependent lower bound that would hold for all strong adaptive variance sequences.
>
> -----

---

> ### Author Response · Authors · 2025-11-17
> **Reply to Review**
>
> ----
>
> **Q6**: Furthermore, the problem is quite complex as the decision sets are also being generated adversarailly (as it adaptive or oblivious?) and the adversary may decide the variance before or after the decision set which seems to affect the bounds. In particular, some intuition on why the weak and strong adversary leads to different rates would be helpful.
>
> **A6**:  As we clarified in A4, our primary aim is to provide a lower bound that holds for any adaptive variance sequence, and the adversary seeks to select a variance sequence that minimizes the regret (or "collaborates" with the learner), thereby breaking the lower bound construction.
>
> The bandit learning algorithm usually takes a trade-off between exploration (trying new actions to gain information) and exploitation (choosing the currently best-known action based on history observation).
>
> The intuition comes from the fact that high variance ($\sigma_k^2$) will have a high impact on exploration—making it hard for the learner to collect reliable information—but will have a low impact on exploitation once the optimal action is well-identified.
>
> * For the strong adversary, the adversary can first observe the decision set ($\mathcal{D}_k$). If the decision set is under-explored and the learner is motivated to collect information, the adversary can strategically select a lower variance level ($\sigma_k^2$), allowing the learner to resolve uncertainty quickly. Conversely, if the learner prefers exploitation, the adversary can select a high variance level, which doesn't hurt the learner much. This flexibility allows the adversary to strategically avoid constructing a hard problem instance, thus breaking the lower bound construction.
>
> * However, for the weak adversary, the timing is inverse. The adversary must first commit to the variance level ($\sigma_k^2$) and then we send a selected decision set ($\mathcal{D}_k$) to the learner. For a high variance level, we can send an under-explored decision set ($\mathcal{D}_k$) to impede the learning process. For a low variance level, we can send a well-explored decision set. This constraint limits the learner’s ability to ease the learning process strategically, allowing the construction of the tight variance-dependent lower bounds.
>
> ------
> **Q7**: The scope of technical novelties of this work also seem limited, and it's a bit unsurprising that general variance sequence-based bounds can be derived using a grouping/peeling argument. Can the authors elaborate on difficulties of the general function approximation setting or discuss wider applicability of techniques introduced here?
>
> **A7**: First, while the peeling method is commonly used for deriving upper bounds in algorithm design and analysis, its application has been limited to that domain. Our work is the first to introduce and apply this method to establish lower bounds, resulting in a tight, variance-aware bound over a general (adaptive) variance sequence.
>
> In addition, our results have direct implications for the general function approximation setting. Since the linear bandit problem with dimension $d$ is a case of the finite Eluder Dimension function class with $d_{elu} = d$, our lower bound directly suggests a corresponding lower bound for bandits with this type of general function approximation.  The extension of our peeling and multi-instance framework to a broader class of general function approximation settings remains a complex challenge and will be a focus of our future work.
>
> Finally, to the best of our knowledge, our result provides the first high-probability lower bound for linear contextual bandits, rather than the standard expected lower bound. The technique of converting an expected lower bound setup to a high-probability guarantee is an independent technical contribution that holds broader interest and can potentially be used to derive high-probability lower bounds for a wider class of problems.
>
> -----

---

> > ### Author Response · Authors · 2025-11-24
> > **Experiement Results**
> >
> > We have run a set of comprehensive experiments to explicitly illustrate the difficulty of our hard-to-learn instance construction derived from the proof of Theorem 4.1 and the full details have been added to Appendix A of the revised paper.
> >
> > ***
> >
> > ### Experimental Setup and Design
> >
> > The experimental setup breaks down a contextual linear bandit problem with total dimension $D=10$ into two orthogonal sub-instances $\mathcal{M}_1$ and $\mathcal{M}_2$ (each dimension $d=5$). The problem follows the construction in the proof of Theorem 4.1 which breaks down the problem into several sub-instances.
> >
> > We consider a prefixed variance sequence over $K=4000$ rounds. The variance sequence is piecewise, defined by an abrupt shift at $2000$ rounds:
> >
> > 1.  Low Variance ($\sigma_1=0.1$): Used in the first $2000$ rounds ($k \le 2000$).
> > 2.  High Variance ($\sigma_2=1.0$): Used in the subsequent $2000$ rounds ($2000 < k \le 4000$).
> >
> > To illustrate the necessity of adaptively allocating different instances to the learner based on the variance level, we consider two scenarios for assigning the sub-instances $\mathcal{M}_1$ and $\mathcal{M}_2$ to the learner:
> >
> > 1.  Piecewise Assignment (Hard-to-Learn): $\mathcal{M}_1$ for $K \le 2000$ then $\mathcal{M}_2$ for $K > 2000$.
> > 2.  Alternating Assignment (Rapidly Switching): Alternating between $\mathcal{M}_1$ and $\mathcal{M}_2$ every round.
> >
> > In each round $k$, the decision set $\mathcal{D}_k$ contains $32$ contexts. The base context entries are drawn i.i.d. from $\mathcal{U}(0, 1)$. This context set is masked such that contexts interact only with $\boldsymbol{\mu}_1$ when $\mathcal{M}_1$ is assigned and only with $\boldsymbol{\mu}_2$ when $\mathcal{M}_2$ is assigned. Crucially this orthogonal masking ensures that information gathered from one sub-instance cannot be transferred or used to estimate the parameter vector of the other sub-instance.
> >
> > ----
> >
> > ### Results and Discussion
> >
> > We evaluate two key algorithms: the OFUL and the Weighted OFUL[2]. Weighted OFUL provides a near-optimal variance-dependent regret guarantee for the linear contextual bandit problem assuming the variance for each round is known to the learner. The table below summarizes the average cumulative regret over 4000 rounds.
> >
> > | Algorithm / Assignment | 500 | 1000 | 1500 | 2000 (Switch) | 2500 | 3000 | 3500 | 4000 |
> > | :--- | :---: | :---: | :---: | :---: | :---: | :---: | :---: | :---: |
> > | OFUL (Piecewise) | 71.2564 | 100.3682 | 120.4346 | 135.9829 | 230.9034 | 269.0856 | 294.4300 | 313.8001 |
> > | OFUL (Alternating) | 102.4081 | 152.4678 | 188.4935 | 216.7144 | 241.3390 | 263.3009 | 282.7492 | 300.6526 |
> > | Weighted OFUL (Piecewise) | 4.7459 | 5.2714 | 5.6159 | 5.8610 | 100.7815 | 138.9637 | 164.3080 | 183.6782 |
> > | Weighted OFUL (Alternating) | 9.0067 | 10.0494 | 10.7135 | 11.2031 | 11.6410 | 12.1044 | 12.5673 | 13.0299 |
> >
> > Analysis:
> >
> > 1.  Standard OFUL (variance-independent) performs poorly across all scenarios as it constructs confidence sets similarly regardless of the variance level failing to exploit the low-noise period.
> > 2.  Weighted OFUL reveals the core tension. In the Piecewise Assignment (Hard Case) it achieves extremely low regret in the low-variance phase ($K \le 2000$ Regret 5.8610). However the subsequent orthogonal switch to the high-variance instance $\mathcal{M}_2$ causes a massive regret spike (reaching 183.6782 by $K=4000$) demonstrating that early learning advantage is completely negated when the second instance must be learned from scratch under high noise.
> > 3.  In contrast the Alternating Assignment allows both instances ($\mathcal{M}_1$ and $\mathcal{M}_2$) to receive low-variance exploration early on. Weighted OFUL successfully utilizes this to construct tight confidence sets for both resulting in the lowest overall cumulative regret (13.0299 by $K=4000$).
> >
> > This result illustrates that the early stage with low variance can significantly speed up the learning process in the exploration stage and lead to a low total regret. The ability to adaptively assign the decision set based on the variance level (as we used in constructing the lower bound) can avoid the early exploration stage having much smaller variance than the later exploitation stage thus successfully circumventing the limitation.
> >
> > [1] Improved algorithms for linear stochastic bandits
> >
> > [2] Nearly minimax optimal reinforcement learning for linear mixture markov decision processes

---

### Comment · Area_Chair_7Q3S · 2025-11-27
**Reminder: Please Discuss**

All Reviewers,

Thank you for your time. As the rebuttal has been available for a while, please engage in discussions with the authors and with one another. There are only a few days left before December 3.

Best,
Area Chair

---

### Meta-Review · Area_Chair_UXSj · 2026-01-06

**Summary:**

The paper studies linear contextual bandits with round-varying noise. It shows matching lower bound when the whole variance sequence is fixed in advance. The paper also provides results in various adversarial settings. The reviewers have following concerns.
1. Technical novelty is limited.
2. No enough motivation.
3. Writing can be improved.

3 out of 4 reviewers gave positive feedback on this paper. Overall, it is a solid theory paper on linear contextual bandits with non-trivial technical contributions.

**Reviewer Concerns:**

I think the concerns are well-addressed by the authors.

**Reviewer Scores:**

2,6,6,6. I think the reviewer giving 2 is likely to raise the score.

---

### Decision · Program_Chairs · 2026-01-26

Accept (Poster)